# Efficient and self-adaptive in-situ learning in multilayer memristor neural networks

Can Li [1], Daniel Belkin[1,2], Yunning Li[1], Peng Yan[1,3], Miao Hu [4,7], Ning Ge[5], Hao Jiang[1], Eric Montgomery[4], Peng Lin[1], Zhongrui Wang[1], Wenhao Song[1], John Paul Strachan[4], Mark Barnell[6], Qing Wu[6], R. Stanley Williams [4], J. Joshua Yang [1] & Qiangfei Xia[1]

Memristors with tunable resistance states are emerging building blocks of artificial neural networks. However, in situ learning on a large-scale multiple-layer memristor network has yet to be demonstrated because of challenges in device property engineering and circuit integration. Here we monolithically integrate hafnium oxide-based memristors with a foundry-made transistor array into a multiple-layer neural network. We experimentally demonstrate in situ learning capability and achieve competitive classification accuracy on a standard machine learning dataset, which further confirms that the training algorithm allows the network to adapt to hardware imperfections. Our simulation using the experimental parameters suggests that a larger network would further increase the classification accuracy. The memristor neural network is a promising hardware platform for artificial intelligence with high speed-energy efficiency.

[1] Department of Electrical and Computer Engineering, University of Massachusetts, Amherst, MA 01003, USA. [2] Swarthmore College, Swarthmore, PA 19081, USA. [3] Wuhan National Laboratory for Optoelectronics, Huazhong University of Science and Technology, Wuhan 430074, China. [4] Hewlett Packard Labs, Hewlett Packard Enterprise, Palo Alto, CA 94304, USA. [5] HP Labs, HP Inc., Palo Alto, CA 94304, USA. [6] Air Force Research Laboratory, Information Directorate, Rome, NY 13441, USA. [7] Present address: Department of Electrical and Computer Engineering, Binghamton University, Binghamton, NY 13902, USA. Correspondence and requests for materials should be addressed to J.J.Y. (email: jjyang@umass.edu) or to Q.X. (email: qxia@umass.edu)

With the introduction of hardware accelerators[1–4] for inference in deep neural networks (DNNs)[5–9], the focus on improving overall energy and time performance for artificial intelligence applications is now on training. One promising approach is in-memory analog computation based on memristor crossbars[10–18], for which simulations have indicated potentially significant speed and power advantages over digital complementary metal-oxide-semiconductor (CMOS)[19–23]. However, experimental demonstrations to date have been limited to discrete devices[24,25] or small arrays and simplified problems[26–31]. Here we report an experimental demonstration of highly efficient in situ learning in a multilayer neural network implemented in a $128 \times 64$ memristor array. The network is trained on 80 000 samples from the Modified National Institute of Standards and Technology (MNIST)[32] handwritten digit database with an online algorithm, after which it correctly classifies 91.71% of 10 000 separate test images. This level of performance is obtained with 11% devices in the crossbar unresponsive to programming pulses and the training algorithm blind to the defectivity, demonstrating the self-adapting capability of the in situ learning to hardware imperfections. Our simulation based on the memristor parameters suggested that the accuracy could be higher than 97% with a larger (e.g., $1024 \times 512$) memristor array. Our results indicate that analog memristor neural networks can achieve accuracy approaching that of state-of-the-art digital CMOS systems with potentially significant improvements in speed-energy efficiency.

Memristors offer excellent size scalability (down to 2 nm)[33], fast switching (faster than 100 ps)[34], and low energy per conductance update (lower than 3 fJ)[34]. Their tunable resistance states can be used both to store information and to perform computation, allowing computing and memory to be integrated in a highly parallel architecture. However, given the level of technology maturity, attempts to implement memristive neural networks have struggled with device non-uniformity, resistance level instability, sneak path currents, and wire resistance, which have limited array sizes and system performance. In particular, learning in memristor neural networks has been hampered by significant statistical variations and fluctuations in programmed conductance states and the lack of linear and symmetric responses to electric pulses[35].

Here we develop a reliable two-pulse conductance programming scheme utilizing on-chip series transistors to address the challenges in memristor conductance programming. This in situ training scheme enables the network to continuously adapt and update its knowledge as more training data become available, which significantly improves accuracy and defect tolerance.

## Results

**Linear and symmetric conductance tuning**. We used recently developed Ta/HfO$_2$/Pt memristors to achieve stable tunable multilevel behavior with a linear current–voltage (IV) relationship[36,37]. The memristors were monolithically integrated with foundry-made transistor arrays on a 6-inch wafer (see Methods). Each memristor was connected to a series transistor in a "1T1R" configuration (Fig. 1a–e shows the integrated memristor array from wafer scale to nanometer scale). To increase the conductance of a given cross point, we applied synchronized positive voltage pulses from a driving circuit board to the memristor top electrode and the gate of the series transistor. The gate voltage, which specifies a compliance current, determines the resulting memristor conductance. We decreased the conductance by first applying a sufficient positive pulse to the memristor bottom electrode to initialize the state, and then used the conductance increase scheme to set the memristor to the desired level

(illustrated in Supplementary Fig. 1). With this scheme, we achieved linear and symmetric conductance increase and decrease with minimal cycle-to-cycle (Fig. 1f, g) and device-to-device (Fig. 1f, h) variations. We were able to set the conductance values across the entire $128 \times 64$ array, except for the stuck devices, with reasonably high accuracy using only two electrical pulses to each memristor (Fig. 1i and Supplementary Fig. 2). The speed and reliability of the conductance update scheme make it possible to train the network in situ with almost any standard algorithm. Here the network was trained using stochastic gradient descent (SGD)[32] to classify handwritten digits in MNIST dataset. For each new sample of training data, the network first performs inference to get the log-probability of the label for each output by the softmax function, and then the weights in each layer are updated accordingly (see Methods).

**In situ training in memristor crossbar**. To implement the SGD algorithm in the memristor crossbar, each synaptic weight was encoded as the difference of the conductance between two memristors. Inference was performed by biasing the top electrodes of memristors in the first layer with a set of voltages whose amplitudes encode an image, then reading the currents from the bottom electrodes of devices in the final layer (Fig. 2a, b) using custom-built circuit boards that can address up to 64 channels in parallel (Supplementary Fig. 3 and Supplementary Fig. 4). During inference, all transistors operate in the deep triode region, and the memristor array becomes a pseudo-crossbar capable of performing matrix multiplication following Ohm's law and Kirchhoff's current law[37,38] (see Supplementary Fig. 5a). The hidden neurons after each layer apply a nonlinear activation (in this work, a rectified linear function in software) to the weighted sums computed in the crossbar. The desired weight update ($\Delta \mathbf{W}$) for each layer was calculated in software using Eq. 1, then applied to the crossbar by the measurement system (see Fig. 2c for the algorithm flow chart, Supplementary Fig. 6 for a unified modeling language class diagram, and Supplementary Fig. 5b, c for a parallel weight update scheme).

$$\Delta \mathbf{W}_l = \eta \cdot \sum_{n=1}^{B} \boldsymbol{\delta}_l(n) \mathbf{v}_l(n)^\top \qquad (1)$$

where $\eta$ is the learning rate, $\mathbf{v}$ is the input voltage column vector for the $l$th layer, $\boldsymbol{\delta}_l$ the output error column vector for the layer, $n$ indexes over the sample, and $B$ is the batch size. For a network with $L$ layers, the error row vectors are computed using

$$\delta_j^l = \begin{cases} y_j - t_j, & l = L; \\ \sum_i w_{ij}^l \delta_i^{l+1}, & l < L \text{ and } I_j > 0; \\ 0, & l < L \text{ and } I_j \leq 0. \end{cases} \qquad (2)$$

where $y_j$ is the Bayesian probability computed by the network and $t_j$ is 1 if this sample belongs to class $j$ and 0 otherwise. This calculation ensures that the network maximizes the log-likelihood of the correct classification for each example.

The error backpropagation[39,40] in this work is calculated in software from the values of the readout weights (see Methods). In the future, backpropagation can be implemented within the memristor crossbar by applying a voltage vector representing the current-layer error to the bottom electrodes of the crossbar and reading out the current vector from the top electrodes for the previous-layer error. An on-chip integrated peripheral for full hardware implemented functionality is under development, which has been discussed and simulated in the literature as well[41–45].

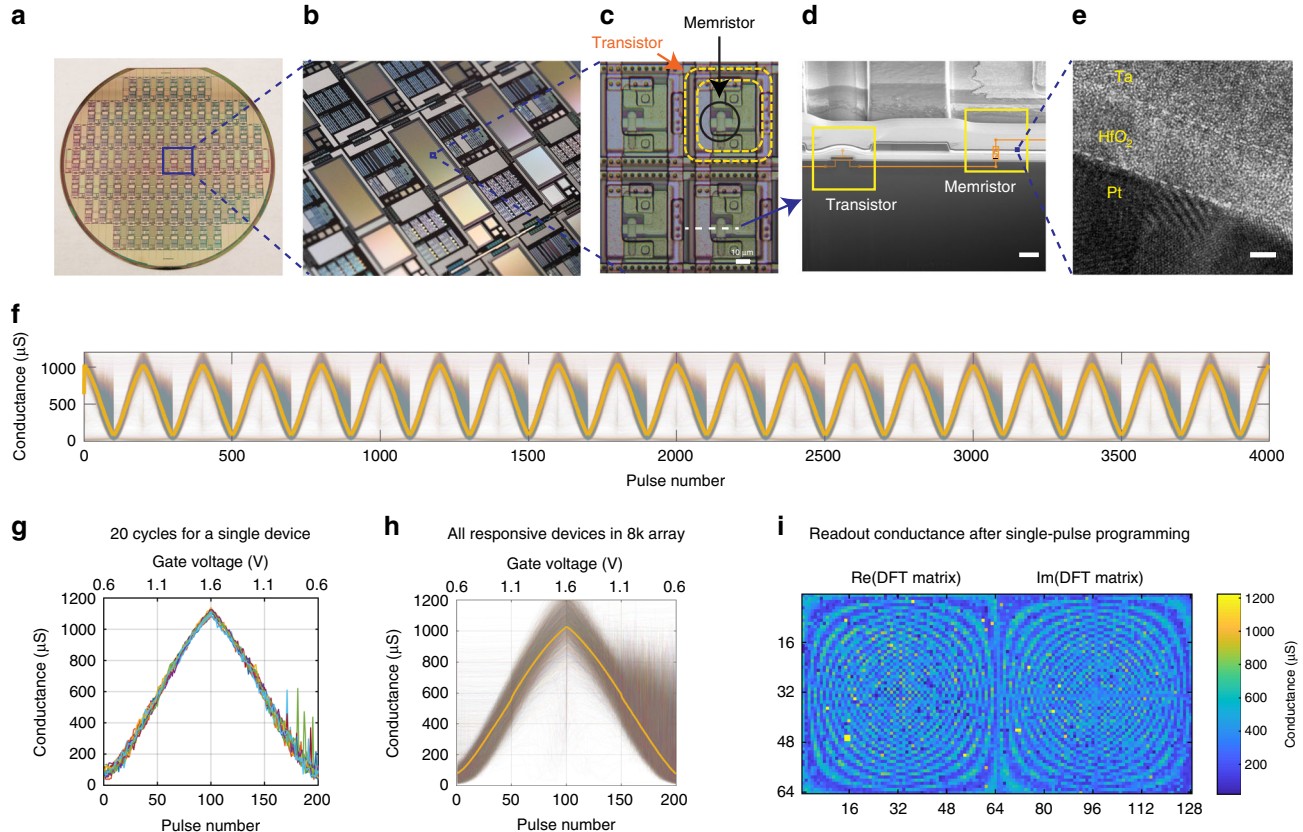

**Fig. 1** Memristive platform for in situ learning. **a** An optical image of a wafer with transistor arrays. **b** Close-up of chip image showing arrays of various sizes. **c** Microscope image showing the 1T1R (one transistor one memristor) structure of the cell. Scale bar, 10 μm. **d** Cross-sectional scanning electron microscopic image of an individual 1T1R cell, which is cut in a focused ion beam microscope from the dashed line in **c**. Scale bar, 2 μm. **e** Cross-sectional transmission electron microscopic image of the integrated Ta/HfO$_2$/Pt memristor. Scale bar, 2 nm. **f** All responsive devices over 20 potentiation/depression epochs of 200 pulses each. **g** Evolution of conductance during 20 cycles of full potentiation and depression for a single cell with 200 pulses per cycle, showing low cycle-to-cycle variability. More results are shown in Supplementary Fig. 1. **h** Evolution of conductance over one 200-pulse cycle of full potentiation and depression for all responsive devices in the array, with median conductance indicated by the yellow line. **i** Conductance of a 128 × 64 array after single-pulse conductance writing of the discrete Fourier transform matrix. Several stuck devices are visible (in yellow)

**Classification of MNIST handwritten digits**. We partitioned a single 128 × 64 array and constructed a two-layer perceptron with 64 input neurons, 54 hidden neurons, and 10 output neurons to be trained on the MNIST dataset of handwritten digits "0" through "9", which has become a standard benchmark by which to gauge new machine learning algorithms. Each input image was rescaled to 8 pixels by 8 pixels (see Supplementary Fig. 7, and sample images in Fig. 3a) to match our network size. The intensities of each pixel of the grayscale images were unrolled into 64-dimensional input feature vectors, which were duplicated to produce 128 analogue voltages to enable negative effective weights (Fig. 2b). The two-layer network used 7992 memristors (see Fig. 3b for the partition on a 128 × 64 array), each of which was initialized with a single pulse with a 1.0 V gate voltage from a low-conductance state. The network was then trained on 80 000 images drawn from the training database (some images were drawn more than once), with a minibatch size $B = 50$ for a total of 1600 training cycles. The smoothed minibatch experimental accuracy (compared with a defect-free simulation) during online training is shown in Fig. 3c. Figure 3d shows the linear relationship between the conductance and the applied gate voltage during each update cycle, which was critical for this demonstration. More analyses are shown in Supplementary Figs. 8, 9. After utilizing the entire training database, the network correctly classified 91.71% of the 10 000 images in the separate test set

(Figs. 3e–j, Supplementary Table 1). Many of the misclassified images are in fact difficult for humans to identify at the available resolution (Fig. 3h, and more in Supplementary Fig. 10).

To understand the potential of the memristor array, we developed a simulation model (see Supplementary Fig. 6 for detailed architecture) based on measured parameters such as the unresponsive rate, conductance update error, and limited memristor conductance dynamic range. We found that the simulated accuracy agrees well with the experimentally achieved one (Fig. 4a), validating the simulation model. A further simulation on a defect-free network shows that the MNIST classification accuracy is similar to that of the same network architecture trained in TensorFlow[46] that uses 32-bit floating point numbers. This result suggests that even though the analogue memristor network has limited precision due to conductance variation, they do not have a significant impact on MNIST classification accuracy (more analysis on conductance update variation is shown in Supplementary Fig. 11). Our finding is consistent with previous theoretical and simulation studies on more sophisticated problems[20,24,47–51]. The analog precision could potentially be improved, if needed for other applications, by device engineering for better IV linearity[37,38], or using pulse width instead of amplitude to represent analog input (with increased time overhead)[27,28,37], or employing multiple memristors to represent one synaptic weight (at the expense of chip area)[52], etc.

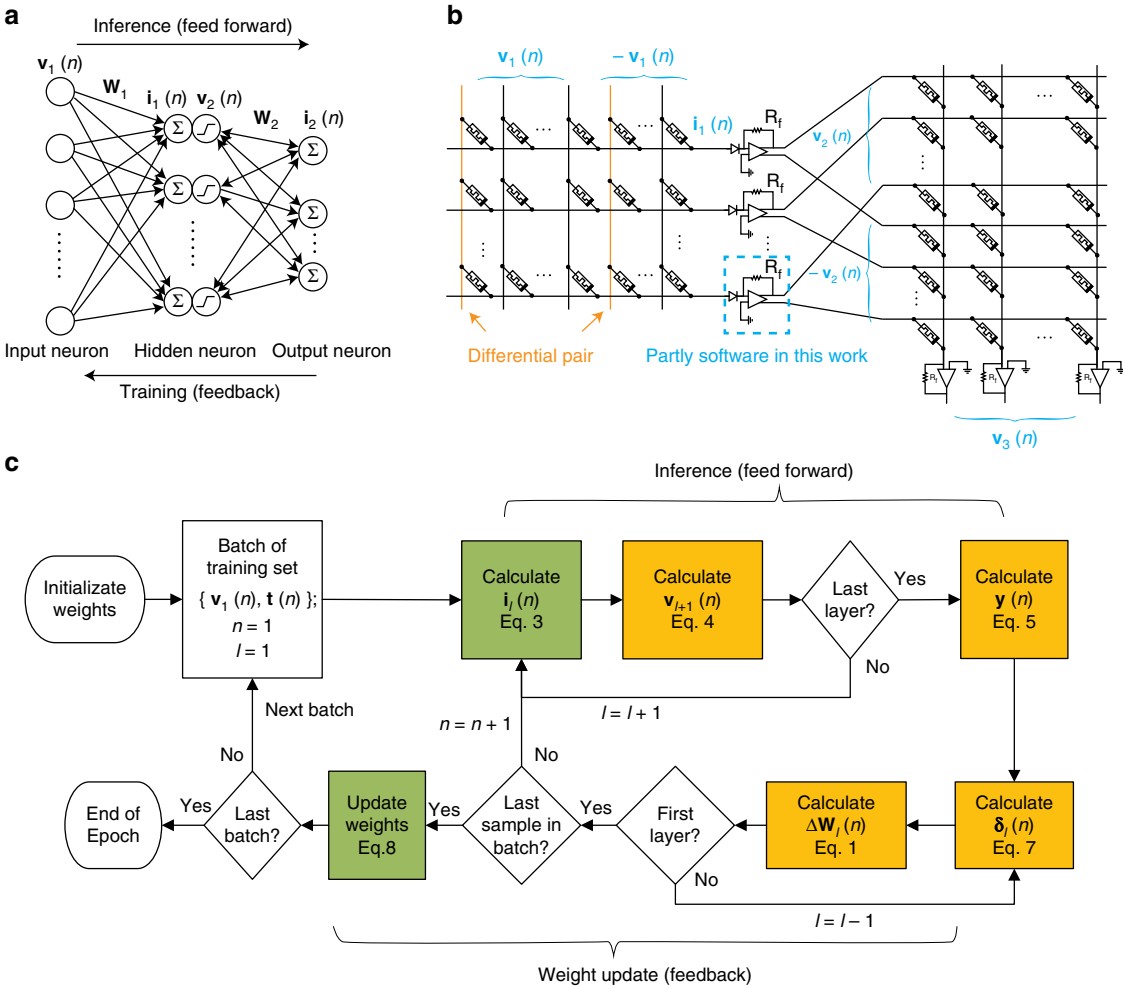

**Fig. 2** In situ training algorithm. **a** Schematic diagram of a two-layer neural network. Each neuron computes a weighted sum of its inputs and applies a nonlinear activation function. **b** The implementation of the network with a set of memristor crossbars. Each synaptic weight (arrows in **a**) corresponds to the conductance difference between two memristors (as illustrated by the orange columns). Each crossbar computes weighted sums of its input voltages. Between the crossbars is a layer of circuits that read the current from each wire, convert it to a voltage, and apply the activation function. The activation function was implemented in software in this work. **c** Flow chart of the in situ training. Steps in green boxes were implemented in hardware in this work, while those in yellow boxes were computationally expensive steps that can be accomplished with circuits integrated onto the chip in the future. The algorithm is described in detail in Methods

A multilayer neural network trained with the online algorithm is more tolerant to hardware imperfections. The experimental accuracy of 91.71% in this work was achieved with 11% devices unresponsive to conductance updates. Our simulation showed that even with 50% of the memristors stuck in a low-conductance state, a >60% classification accuracy is still possible through online training (Fig. 4b), although the accuracy is much more sensitive to shorted devices (Supplementary Fig. 12). On the other hand, if pre-trained weights are loaded to the memristor crossbar (i.e., ex situ training), the classification accuracy decreases quickly with the defect rate (Fig. 4b)). There are approaches to improve the robustness of ex situ training[19,38], but most of them require that the parameters be tuned based on specific knowledge of the hardware (e.g., peripheral circuitry) and memristor array (e.g., device defects, wire resistance, etc.), while the in situ training adapts the weights and compensates them automatically. The self-adaption is more powerful in a deeper network in which the hidden neurons are able to minimize the impact of defects, as suggested by the higher classification accuracy from a two-layer network than that from a single-layer one on the same images (Fig. 4c). The online training is also able to update the weights to compensate for possible hardware and memristor conductance drift over time (see Supplementary Fig. 13).

We expect that the classification accuracy can be improved substantially with a larger network that has more hidden units and/or more inputs to support images with higher resolution. We performed a simulation with our model on a $1024 \times 512$ memristor array, which is likely to be available in the near future, to recognize images of $22 \times 22$ pixels cropped from the MNIST dataset. The network consists of 484 input neurons, 502 hidden neurons, 10 output neurons, and a total of 495 976 memristors in the two layers to represent the synaptic weights (see Supplementary Fig. 14). After training on 1,200,000 images (20 epochs), a $97.3 \pm 0.4\%$ classification accuracy is achieved on the test set even with 11% stuck devices, approaching that demonstrated with traditional hardware. It will be straightforward to build deeper fully connected neural networks on an integrated chip with multiple large arrays in the near future, for even better accuracy and application to more complicated tasks. It is also noteworthy that most state-of-art DNNs involve sophisticated microstructures, e.g., convolutional neural networks (CNNs) or long short-term memory units (LSTMs). It may be worth

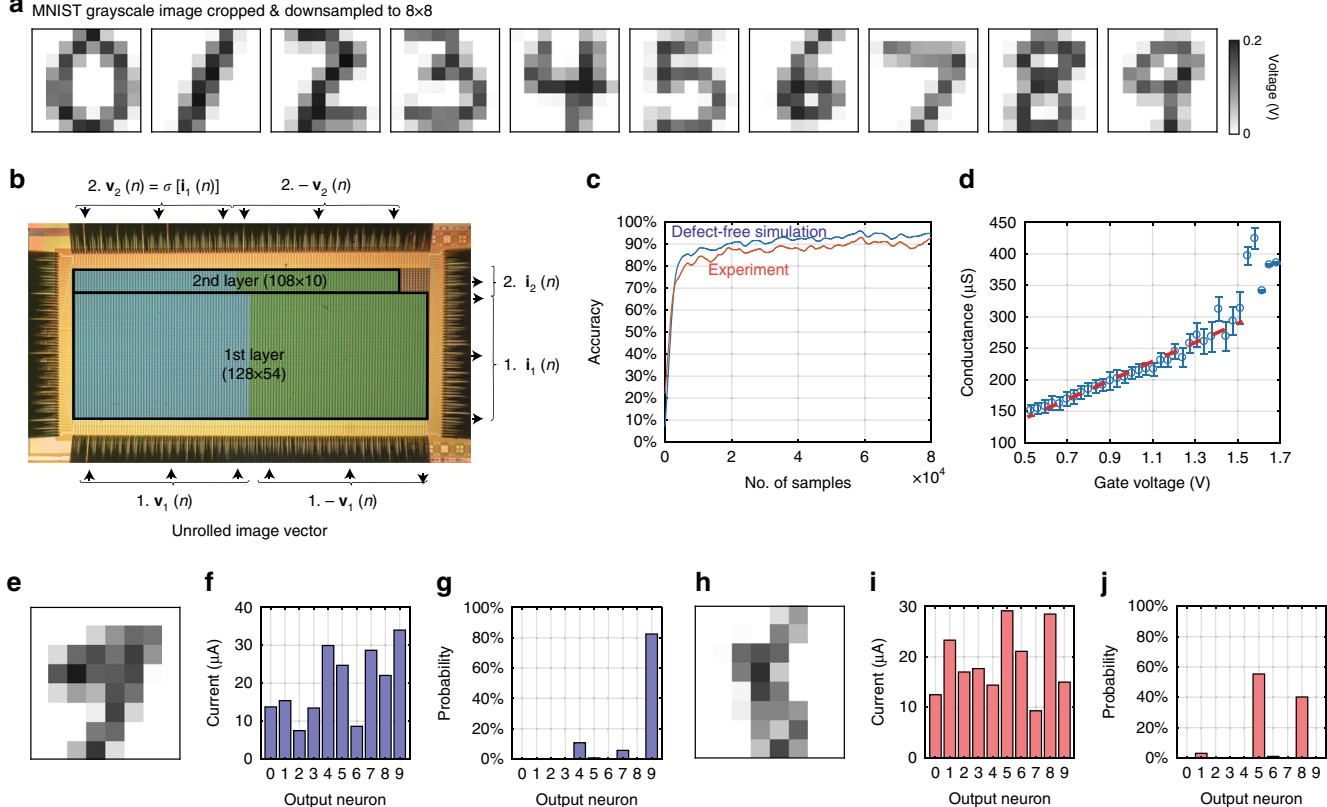

**Fig. 3** In situ online training and inference experiments on Modified National Institute of Standards and Technology (MNIST) handwritten digit recognition. **a** Typical handwritten digits from the MNIST database. **b** Photo of the integrated 128 × 64 array during measurement. The array was partitioned into two parts for the first and second layers, respectively. In all, 54 hidden neurons were used, so the first layer weight matrix is 64 × 54 (implemented using 6912 memristors) and the second layer matrix is 54 × 10 (implemented using 1080 memristors). The blue and green false-colored areas are the positive and negative parts of the differential pairs. **c** Minibatch accuracy increases over the course of training. Experimental data followed the defect-free simulation closely, with a consistent 2–4% gap. **d** The conductance-gate voltage relation extracted from data collected during training. The conductance was read using the scheme described in the Methods. The conductance includes the effects of sneak-paths and wire resistance, which makes the measured values smaller and the variance larger than those in Fig. 1b, c. The dashed line indicates the mean conductance, while the error bars show a 95% confidence interval for the measured conductance. The real-time online training accuracy with the readout weight values is shown in an animation in Supplementary Movie 1. **e**–**g** Typical correctly classified digit "9" and **h**–**j** misclassified digit "8" after the in situ training. **e**, **h** Images of the actual digits from the MNIST test set used as the input to the network. **f**, **i** The raw current measured from the output layer neurons. The neuron representing the digit "9" has the highest output current, indicating a correct classification. **g**, **j** The corresponding Bayesian probability of each digit, as calculated by a softmax function. More inference samples are shown in Supplementary Fig. 7 and Supplementary Movies 2 and 3

investigating how to implement CNNs[37] or LSTMs efficiently on memristor crossbars in the future. But on the other hand, such microstructure-based algorithms have been developed for use on conventional hardware, on which it is more efficient to process sparse matrices. Since the advantages of using sparse matrices in a memristor crossbar are minimal, the optimal architectures for sophisticated tasks may look different.

## Discussion

A further potential benefit of utilizing analog computation in a memristor-based neural network is a substantial improvement in speed-energy efficiency. The advantages mainly come from the fact that the computation is performed in the same location used to store the network data, which minimizes the time and energy cost of accessing the network parameters required by the conventional von-Neumann architecture. The analog memristor network is also capable of handling analog data acquired directly from sensors, which further reduces the energy overhead from analog-to-digital conversion. The memristors we used maintain a highly linear IV relationship, allowing for the use of

voltage-amplitude as the analog input for each layer. This also minimizes circuit complexity and hence energy consumption for future hardware hidden neurons and output current readout. While the external control electronics we use in this work is not optimized for fast speed and low power consumption yet, previous literature on circuit design[45,51] and architecture[21,53] suggest an on-chip integrated system would yield significant advantages in speed-energy efficiency.

In summary, we have demonstrated the in situ and self-adaptive learning capability of a multilayer neural network built by monolithically integrating memristor arrays onto a foundry-made CMOS substrate. The transistors enable reliable, linear, and symmetric synaptic weight updates, allowing the network to be trained with standard machine learning algorithms. After training with a SGD algorithm on 80 000 images drawn from the MNIST training set, we achieved 91.71% accuracy on the complete 10,000-image test set. This accuracy is 2.4% lower than an idealized simulation despite an 11% defect rate for the memristors used. The demonstrated performance with in situ online training and inference suggests that memristor crossbars are a promising high speed and energy efficiency technology for artificial

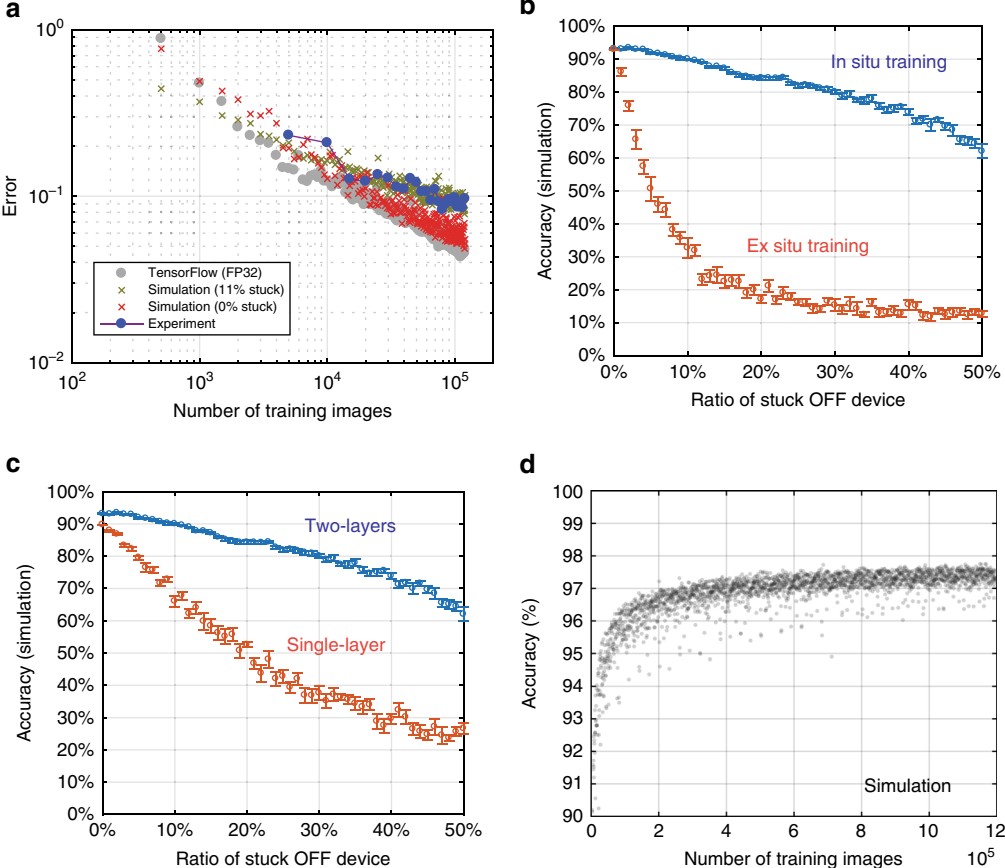

**Fig. 4** Analysis based on experimental-calibrated simulation. **a** The experimental classification error (accuracy shown in Supplementary Fig. 10) matches the simulated accuracy. The simulation considers experiment parameters, including 11% devices stuck at 10 µS, 2% conductance update variation, limited conductance dynamic range, etc. The simulation on defect-free assumption shows an accuracy approaching that from TensorFlow. Each data point is the classification error rate on the complete testing set (10 000 images) after 500 images (simulation or TensorFlow) or 5000 images (experiment). **b** The impact of non-responsive devices on the inference accuracy with in situ and ex situ training approach. The non-responsive device was stuck in a very low-conductance state (10 µS), which is the typical defect device value observed in the experiment. The result shows that the in situ training process adapts to the defects, providing a much higher defect tolerance compared with pre-loading ex situ training weights into the network. With 50% stuck OFF devices, the network can still achieve over 60% accuracy. The error bar shows the s.d. over 10 simulations. **c** The multilayer network also helps with the defect tolerance. If one device is stuck, the associated hidden neuron will adjust the connections accordingly. The error bar shows the s.d. over 10 simulations. **d** The simulation of a larger network constructed on a larger memristor crossbar (1024 × 512) with experimental parameters (e.g., 11% defect rate) could achieve accuracy above 97%, which suggests a large memristor network could narrow the accuracy performance gap from the conventional CMOS hardware. The network architecture is shown in Supplementary Fig. 14

intelligence applications. The software neurons used in this demonstration indicate that a hybrid digital processor and neuromorphic analogue approach for DNNs can be effective, but all the software functions used in the present demonstration can be integrated as hardware onto a full-function chip in the near future.

## Methods

**Device fabrication and array integration**. The transistor array and interconnection between cells are taped out from a commercial foundry with 2 µm technology node to achieve low wire resistance. We monolithically integrate memristors on top of as-received chip in house. Pd/Ag contact metals are first deposited on both vias after argon plasma treatment to remove the native oxide. The chip is then annealed at 300 °C for 30 min in 20 s.c.c.m nitrogen flow to achieve good electrical contact. The memristor bottom electrode is deposited by evaporating 20 nm Pt on top of a 2 nm-thick Ta adhesive layer and patterned by photolithography and lift-off in acetone. A switching layer of 5 nm-thick $HfO_2$ is deposited by atomic layer deposition using water and tetrakis (dimethylamido)hafnium as precursors at 250 °C, and then patterned by photolithography and reactive ion etch. Finally, the top electrode of 50 nm-thick Ta is deposited by sputtering and lift-off, followed by sputtering of a 10 nm-thick Pd protection layer.

**Dataset**. The dataset is composed of the input feature vector ($\mathbf{x}(n)$ for sample $n$) and the target output vector ($\mathbf{t}(n)$). For the classification problem, $t_c(n) = 1$ if sample $n$ belongs to class $c$, and is 0 otherwise. For the MNIST dataset, feature vectors are the unrolled grayscale pixel values of the handwritten digital two-dimensional images. The original images are 28 pixels by 28 pixels. They were cropped to 20 × 20 and then further down sampled to 8 × 8 (using bicubic interpolation) to match the input size of the memristor neural network (Supplementary Fig. 7). The 8 × 8 grayscale images were then unrolled to 64-dimensional input feature vectors. The input feature vectors were converted to voltage values $\mathbf{v}_1(n)$ by a scaling factor, which was the same for all images. The output vectors have 10 dimensions, each corresponding to one digit.

**Inference**. The in situ online training was composed of two stages: feedforward inference and feedback weight update. The multilayer inference was performed layer by layer sequentially. The input voltage vector to the first layer was a feature vector from the dataset, while the input vector for the subsequent layer was the output vector of the previous layer. The analog weighted sum step was performed in the memristor crossbar array as indicated by Eq. 3, or equivalently by Eq. 4:

$$\mathbf{i}_l = \mathbf{W}_l \mathbf{v}_l \qquad (3)$$

$$I_j = \sum_{i=1}^{n} w_{ij} \cdot V_i = \sum_{i=1}^{n} \left[ G_{ij}^{+} \cdot V_i + G_{ij}^{-} \cdot (-V_i) \right] \qquad (4)$$

where $\mathbf{v}_l$ is the $l$th layer input voltage vector that is applied to the top electrodes of the memristor crossbar, $\mathbf{i}_l$ is the readout current vector from the bottom electrodes of the crossbar, and $\mathbf{W}_l$ is the weight matrix of layer $l$. The total current is the sum of the currents through each device in the same column (Kirchhoff's current law), while each current is the product of the conductance and the voltage across the memristor (Ohm's law). Each weight value is represented by the difference in conductance between two memristors: $W_{ij} = G_{ij}^+ - G_{ij}^-$, so that weights can be negative. This is accomplished by duplicating the voltage vector $\mathbf{v}_l$, with $+\mathbf{v}_l$ applied to half of the array and $-\mathbf{v}_l$ applied to the other half, as shown in Fig. 2c.

We chose a rectified linear unit activation function for the hidden layer, which is defined in

$$V_i^{l+1} = \sigma(I_i^l) = \begin{cases} cI_i^l, & I_i^l > 0; \\ 0, & I_i^l \leq 0. \end{cases} \tag{5}$$

where $c$ is a scaling factor to match the voltage range. For the MNIST network in this work, $c$ was set to 200 V/A, and elements of the resulting voltage vector that exceed 0.2 V were clipped to avoid altering the memristor states. This particular step was implemented by software in this investigation, and can be easily implemented in the future with a rectifying diode and amplifier on an integrated chip. The most active (highest amplitude) output current was interpreted as the classification result.

**Softmax and cross-entropy loss function**. The inference result can also be calculated as a Bayesian probability, using the conversion defined by Eq. 6.

$$y_c(n) = \frac{e^{kI_c(n)}}{\sum_{m=1}^{C} e^{kI_m(n)}} \tag{6}$$

where $y_c(n)$ is the probability that sample $n$ belongs to class $c$, and $C$ is the total number of classes. $k$ was set to $5 \times 10^5$/A for the MNIST network in this work.

The goal of the training process was to adjust the weight values to maximize the log-likelihood of the true class. We used a cross-entropy loss function, which is defined in Eq. 7.

$$\xi(T, Y) = \sum_{n=1}^{N} \xi[t(n), y(n)] = -\sum_{n=1}^{N} \sum_{c=1}^{C} t_c(n) \log[y_c(n)] \tag{7}$$

where $N$ is the total number of samples.

**SGD with backpropagation**. In order to estimate the gradient of the loss function for training the weights, we withdrew a subset of $B$ samples (termed a minibatch) from the training set without replacement at each round of training. The SGD algorithm was used to update the weights along the direction of steepest descent for $\mathbb{E}[\xi]$. The desired weight update was given by Eq. 1 in the main text. For a network with $L$ layers, the error vector is computed by

$$\delta_j^l = \begin{cases} \frac{\partial \xi}{\partial I_j^l}, & l = L; \\ \frac{\partial \sigma}{\partial I_j^l} \sum_i W_{ij}^l \delta_i^{l+1}, & l \leq L. \end{cases} \tag{8}$$

where $\sigma$ is the nonlinear activation function for the hidden layer and $\xi$ the loss function of the output layer. For the loss function utilized in this study and rectified linear activation, Eq. 8 reduces to Eq. 2 in the main text and was evaluated in software. With some improvements to the measurement system, we will be able to implement this step in the crossbar, as described in the main text.

The weight update is then applied to the crossbar. We first adjust the gate voltages of the transistors following Eq. 9. The changes in the gate voltage for the memristors in differential pairs are of the same magnitude but in opposite directions. We enforced a maximum and a minimum gate voltage of 1.7 and 0.6 V, respectively, for the current chip to make sure the transistor-gate-voltage and memristor conductance relation remains in the linear region.

$$\Delta \mathbf{V}_{\text{gate},l} = [+\Delta \mathbf{W}_l - \Delta \mathbf{W}_l] \tag{9}$$

For memristors for which $\Delta V_{\text{gate},l,ij} < 0$, we first apply a voltage pulse (1.6 V) on the bottom electrodes of the array to initialize the memristor to a very low-conductance state. We then apply a voltage pulse (2.5 V) to the top electrodes with an updated voltage matrix applied to the gates of the series transistors, which raises the memristor conductance state up to match the gate voltage. As shown in the main text, the resulting memristor conductance depends linearly on the transistor's gate voltage during this update process (Fig. 1f).

**Reading the conductance matrix**. To read the conductance of cross point $(i, j)$ directly, we turn off all transistors except for column $j$, which is left fully on, then apply a voltage to row $i$ and ground all other rows. The current out of column $j$ is read, and we use the relation

$$V_i = I_j \left[ R_{ij} + R_w(i,j) \right] \tag{10}$$

where $R_w(i,j)$ is the total wire resistance along the unique conductive path. For a known wire resistance per segment $R_s$ in an $N \times M$ array with voltages applied from the left ($j = 0$) edge and the lower ($i = N$) edge grounded, we calculate

$$R_w(i,j) = R_s(N + 2 + j - i) \tag{11}$$

Together, these equations give

$$G_{ij} = \frac{1}{\frac{V_i}{I_j} - R_s(N + 2 + j - i)} \tag{12}$$

as the exact conductance of the memristor itself. However, because of wire resistance and the sneak path problem, this conductance cannot be used directly to predict the behavior of the array during vector-matrix multiplication operations. Also, each memristor must be read sequentially, so the time complexity of this approach is proportional to $NM$.

For any linear physical system with $N$ inputs and $M$ outputs, there is an equivalent linear transformation implemented by the system that can be represented as an $N \times M$ matrix. We can use this fact to read the array more efficiently. In particular, for any possible network of Ohmic resistors with $N$ voltage inputs and $M$ current outputs, there is a matrix $\mathbf{G_{eq}}$ such that for any matrix $\mathbf{V}$ in $\mathbb{R}^{N \times P}$, $\mathbf{I} = \mathbf{G_{eq}} \mathbf{V}$. Because the IV relationship in the Ta/HfO$_2$/Pt memristor is highly linear, such a matrix exists for our array, which is the equivalent conductance matrix. This matrix is electrically indistinguishable from our physical array (to the extent that the components used are linear). We can determine $\mathbf{G_{eq}}$ empirically by running matrix-matrix multiplication with some large matrix of inputs $\mathbf{V}$ and using the equation $\mathbf{G_{eq}} = \mathbf{IV}^{-1}$. In practice, we usually choose $\mathbf{V}$ to be the $N \times N$ identity matrix. The runtime of this calculation is proportional to $N$.

**Data availability**. The data that support the findings of this study are available from the corresponding author upon request.

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

## Acknowledgements

This work was supported in part by the Air Force Research Laboratory (AFRL) (Grant No. FA8750-15-2-0044) and the Intelligence Advanced Research Projects Activity (IARPA) (contract 2014-14080800008). D.B. is supported by a Research Experience for Undergraduates (REU) supplement grant from NSF (ECCS-1253073). P.Y. acknowledges the support from the Chinese Scholarship Council (CSC) under grant 201606160074. This work was performed in part at the Center for Hierarchical Manufacturing (CHM), an NSF sponsored Nanoscale Science and Engineering Center (NSEC) at University of Massachusetts, Amherst, and in part at the Center for Nanoscale Systems (CNS), a member of the NSF National Nanotechnology Infrastructure Network (NNIN) at Harvard University (ECS-0335765).

## Author contributions

C.L. and D.B. did the programming, measurements, data analysis, and simulation. C.L., P.Y., N.G., H.J., and P.L., built the integrated chips. Y.L., C.L., W.S., M.H., E.M., Z.W., and J.P.S. built the measurement system and firmware. C.L. and H.J. took the SEM and TEM images. Q.X. and J.J.Y. designed and supervised the project. Q.X., C.L., D.B., R.S.W., and J.J.Y. wrote the manuscript. M.B. and Q.W. and all other authors contributed to the result analysis and commented on the manuscript.

## Additional information

**Competing interests:** The authors declare no competing interests.

