## [Peer Review File · Nature Communications]

Reviewers' comments:

Reviewer #1 (Remarks to the Author):

The article "Efficient and self-adaptive in-situ learning in multilayer memristor neural networks" is a compelling illustration of one of the most elusive challenges to using memristor technology in machine learning applications – getting learning "on the chip".

This is an important finding, and one that the authors should be commended for.

That said, the authors tend to rely on scientific exaggeration in several areas of this manuscript; somewhat unnecessarily so. This both serves to inappropriately emphasize the maturity of these findings to those unfamiliar with this field and will be offputting to others in the field; and in the end this will diminish the impact of this paper.

Specifically with regard to the strength of the claims:

1) The title says "self-adaptive." While I'm not sure what the authors specifically mean by this term, the system described in the paper is clearly not stand-alone, requiring many of the computations to be performed off the memristor chip in software. That these steps could in principle be hardware accelerated with simple circuits is good to point out, but the system being described in this paper does not appear to be stand-alone.

At minimum, the authors should be explicit about what "self-adaptive" means.

2) The abstract and subsequent descriptions suggest that they are performing "Bayesian Inference." While in a broad sense of the word this could be said to be the case (most neural networks can be described through a Bayesian perspective), the authors appear to be simply implementing a standard 3 layer neural network (or 2 layers, depending how you count) with a software log-likelihood softmax as the output and input to the training signal. It isn't immediately obvious that this analysis is strictly Bayesian, aside from their probabilistic normalization in equation 5. At least the way described in the paper, it comes off as standard neural network training with a fancy name. If the authors have a stronger Bayesian rationale or justification, they should be explicit about it.

3) 91.71% accuracy on MNIST is fairly horrible by computer vision standards. Yes, the images were reduced in size to 8x8 to fit on the crossbar; but a generic machine learning researcher will see

that number and dismiss the paper. The authors need to acknowledge this discrepancy and address how larger hardware will likely do better.

a. Related, there is a clear precision concern around memristors and similar devices for analog computing of neural networks. There is a growing literature around this subject (hardware papers by Burr and Agarwal and others; machine learning papers by Bengio and others regarding low-precision networks). The authors should include deeper discussions about precision and machine learning

4) The energy analysis in both the paper and methods reads a bit like a sales brochure. First, I don't understand why they are comparing the cost of an analog operation to a FLOP, when clearly the analog operation is not at the same precision. Whatever alteration (more memristors?) has to be made to the circuit to achieve comparable digital precision will likely increase cost. There is a literature around this precision consideration that the authors should cite. Further, the conventional CMOS operations are generic, so the extra circuitry that seems to be ignored in the analysis of their own system probably should be accounted for as well (if it is not already, it was not completely clear). I'm also not entirely sure, but is not the number quoted for TrueNorth (>20pJ per operation) for spike events? The whole point of TrueNorth is that it is event driven, thus proportionally a lot more computation occurs with each spike than in a conventional operation.

Again, the value of analog computing is well established if systems can be efficiently trained and scaled using the technology described here. The authors do not need to use poorly set up apples-to-oranges comparisons to make their technology look better than competitors. At least not in an academic publication.

5) The time analysis suffers from a similar problem. They treat the crossbar as a parallel architecture uniquely, but of course GPUs and CPUs can also be parallelized. The low-level mat-vec kernels are not as easy to split up in conventional treatments, but training batches are routinely distributed over many cores and results then brought back together. It is a different parallel model to be sure, but to say that a GPU can be matched by a 512x256 array ignores the fact that I could theoretically string together many GPUs to parallelize my training over a very large data set, which is not that immediately feasible for the analog system.

a. Related, most state-of-the-art neural networks are convolutional (at least in part) or involve more sophisticated micro-networks, like LSTMs. The simplest way to think of these for the analysis here is that these algorithms leverage sparse matrices; which presumably the conventional hardware can take advantage of, while the physical crossbars would benefit minimally from (at least in terms of time).

All of these points are not to question the overall value of the study. Clearly, the efficient training of analog memristor accelerators for neural networks is a huge and important win. But the authors do not need to oversell this by framing comparative analyses in such a way as to make best case assumptions for their own technology and worst case for alternatives. It is kind of like putting cheap

chrome on a Ferrari. From a distance it may look impressive, but up close it is a waste and distraction.

Some minor questions / points:

6) Of the 36 citations, the vast majority are to memristor papers, with 28 in the first three lines of the abstract; whereas there are almost no citations for the algorithm they're using and minimal to the broader machine learning applications (ignoring the growing literature around low-precision ANNs) that could be impacted. For a Nature Communications article, I'd suggest a subset of references suitable for a broad audience, not a narrow community of device researchers. I'd restrict the memristor citations to those that are specifically relevant for the work being described here.

7) On the algorithm implementation (Figure 2), how are the errors from each sample accumulated or stored prior to the update weights (once per batch) on hardware? This accumulation is in software, right?

8) The ideal algorithm is the exact same network from layer 2A, but performed in floating point in simulation? Or with approximate analog precision minus the defect noise? Did the authors try a state-of-the-art computer vision algorithm (e.g., deep convolutional net) on the lower-resolution images to quantify the impact of the low-res?

9) I'm not sure the confusion matrix (table 1) adds anything here, though it may be interesting if the software-"ideal" case yielded a qualitatively different confusion matrix

10) Top of page 10, the authors state "an advantage of online training is the ability to update the weights continuously during use, to compensate for conductance drift over time and changes in the input data distribution". This is true; however, the authors should point out that neural networks suitable for transfer learning / adapting to concept drift are still very preliminary.

The case they point out about using this online training to compensate for hardware drift is a very important observation, and probably merits more discussion.

11) Page 11 (And in methods) As mentioned above, TrueNorth's power consumption is ~22pJ per spike event, not per algorithmic operation (the event-driven nature of the chip means that they pay ~0pJ when nothing is communicated). This is a subtle but very important distinction – it shifts the burden of energy use onto the algorithm; but it is not fair to imply that TrueNorth is 1000 times more power hungry per operation.

12) It is probably more fair to communicate "error rate" on MNIST as opposed to "accuracy"

Reviewer #2 (Remarks to the Author):

The paper presents a memristive neural network with linear relation between the voltage and the memristor conductance and therefore it can be trained in-situ reliably.

Overall, the paper is well-written and the results are interesting and important. The reviewer thinks that this work is with high quality and deserves publication.

The reviewer feels that there is too much emphasize on the neural network theory rather than focusing on the contributions of the paper. The authors should focus more about the fabrication and process of the memristor array and PCB in use since the design and structure are not clearly described. A schematic of the 1T1R arrays should be provided, including showing how the same memristive array is used for different structures of neural networks (how a 128x64 array becomes 64x54x10 two layer MNN?). Fig. 2 shows pure crossbars that are very different than the real architecture used in this work. Fig. 5 in supp. also seems to present only part of the story. Also the linear programming approach should be better explained. Specifically, how many levels can be achieved? (from Fig. 1, ~100 levels are shown).

It will be worthwhile to provide the code for the software implementation of the training and the missing circuit units.

The presented learning scheme with fixed pulses across the memristor and with varying pulse magnitude on the gate of the transistor is similar to a stochastic gradient descent training technique presented in Soudry et al., " Memristor-based Multilayer Neural Networks with Online Gradient Descent Training," TNNLS 2015. It worth mentioning the differences and similarities. They also provided the circuits for the training in Rosenthal et al., "A Fully Analog Memristor-Based Multilayer Neural Network with Online Backpropagation Training", ISCAS 2016.

While some of the results are trivial (the influence of stuck at ON is more significant than stuck at OFF for example is intuitive), others are interesting. The authors should emphasize what are the important insights and what is only a demonstration of known insights.

The comparison to ex-situ can be explained better. Also, note that there are work arounds that can improve the vulnerability of ex-situ training to the memristive system faults.

Method 6- "With integrated hardware neurons, we estimate that the time needed for each layer of neurons to stabilize can be brought as low as 10 ns" need to justify this claim.

Method 6+7- comparison to execution time of TPU and GPU, increasing the size of the layers till you meet the point where the suggested work outperform GPU\TPU is “artificial”, consider checking the performance for well-known networks such as AlexNet, VGG, ResNet (even only- compare execution of “partial” net).

The energy comparison is not convincing. It seems that the authors ignore the energy of the periphery including costly ADCs and voltage/current sources. A real comparison would be comparing the energy per image, but it requires the entire implementations.

Reviewer #3 (Remarks to the Author):

The paper “Efficient and self-adaptive in-situ learning in multilayer memristor neural networks” by Can Li et al. shows a backpropagation implementation on software-hardware implementation. Although interesting, the paper does not show enough novelty to be published in Nature Communications.

- 1- Please clearly state in the main paper the device programming speed (500us and 5us, as I believe is described in the Supplementary Info), which definitely seems very long for promising very fast training speed.
- 2- In the entire paper, many crucial aspects are explained by saying that there will be efficient circuitry able to perform such aspects. Authors should explain how they actually think to implement such aspects, if A/D D/A converters are used, if operation is performed in parallel or serially, since using such high-density arrays pose severe silicon area constraints, and therefore implementable CMOS functionalities.
- 3- The training is performed using mini-batches. How can the different delta and x values be memorized, before applying the weight update? Also Fig. 2c does not mention this. Where these values will be memorized in circuitry?
- 4- Weight update is performed row by row in the scheme in Fig. S5, which slows down programming operation by orders of magnitude (considering the worst case of 500us per line with a 1024x1024 array, this means 500ms, definitely not useful). In addition, Fig. S1 shows that different conductance levels are obtained by programming different gate voltages, which means that the circuit overhead is unacceptably large.

- 5- I don't understand how the programming scheme works: if I program a device into a particular conductance, and then, i.e., I ask to reduce it, do I need to read the device conductance between the two operations to calculate the exact voltage pulse to be applied?
- 6- Regarding the obtained accuracy of 91.71% (which actually shows values fluctuating below 90% in Fig. 3c), how it is compared with an equivalent size software network? Because the Ideal Software simulation in Fig. 3c is misleading, since it corresponds to a memristor network with no dead or stuck-on devices, not a software (i.e. Tensorflow) simulation.
- 7- The conductance values shown are very large, hundreds or thousands of μS . This leads to a huge power consumption during programming operation. Did authors consider this in their power estimates?
- 8- Please clearly state that Fig. 4 are simulations. What is the real random variation during programming operation of real memristors used in this paper? I think is much more than the optimum 1%.
- 9- Calculations for power consumption completely neglect CMOS circuitry, which can be a large error in case A/D D/A converters are used. Calculations should be redone considering circuitry. In addition, here memristor power consumption is not based on the memristors of the papers, but on the best obtained in the literature. What is the linearity of this cited device, which is the main advantage of the present manuscript memristor?
- 10- How is the 20ns estimate calculated for read time for CMOS circuitry?

We thank all the reviewers for their precious time and constructive comments on our manuscript. We revised both the main text and the supplementary information (SI) accordingly. The changes we made are summarized as follows:

1. Removed quantitative comparison on power consumption between different architectures on page 13–14 of the revised main text.
2. Added comparison between our experimental data, simulation data and software data trained with *TensorFlow*, and projected that a larger network would lead to classification accuracy over 97%. The new paragraphs are added on pages 10–12 in the revised main text and the network is shown in *Supplementary Fig. 14* in the revised SI.
3. Added more analyses on the automatic compensation/adaption of hardware drift by the *in-situ* training, as shown in the *Supplementary Figure 13* in the revised SI.
4. Added details about device fabrication as *Method* on Page 15 of the revised main text; PCB board design as *Supplementary Figure 4* in the revised SI.
5. Added code snippet showing the high-level training algorithm as *Supplementary Note 1* in the revised SI.

In the following point-to-point response, the original reviewers' comments are in black fonts, our responses are in blue fonts. Changes in the revised main text and SI are also highlighted in blue colored italic font.

Response to Reviewers' comments:

Reviewer #1 (Remarks to the Author):

The article “Efficient and self-adaptive in-situ learning in multilayer memristor neural networks” is a compelling illustration of one of the most elusive challenges to using memristor technology in machine learning applications – getting learning “on the chip”.

This is an important finding, and one that the authors should be commended for.

That said, the authors tend to rely on scientific exaggeration in several areas of this manuscript; somewhat unnecessarily so. This both serves to inappropriately emphasize the maturity of these findings to those unfamiliar with this field and will be off putting to others in the field; and in the end this will diminish the impact of this paper.

Response: We thank the reviewer for commending our contribution and pointing out the significance of our experimental demonstration of the “on the chip” learning with memristor technology. We also very much appreciate the reviewer’s suggestions to improve our manuscript by making more objective statements.

Specifically, with regard to the strength of the claims:

1) The title says “self-adaptive.” While I’m not sure what the authors specifically mean by this term, the system described in the paper is clearly not stand-alone, requiring many of the computations to be performed off the memristor chip in software. That these steps could in principle be hardware accelerated with simple circuits is good to point out, but the system being described in this paper does not appear to be stand-alone.

At minimum, the authors should be explicit about what “self-adaptive” means.

Response: We thank the reviewer for this comment. It is true that our current hardware is not a standalone system. The memristor neural network is supported by peripheral driving/control circuits and software neurons. By stating “self-adaptive” we mean that no prior knowledge of the hardware imperfections (e.g., memristor defect, peripheral asymmetry, etc.) is required during the on-line training process. The algorithm (in this case, SGD) corrects the hardware imperfections, or in other words, “self-adapts” to them. This is actually an important point with significant implications. As our experiment results suggest, even if an emerging device (such as a metal-oxide memristor) is not optimized for digital/memory applications, it still can be a good candidate for constructing analog neural networks.

To make this point clearer, we made the following revisions in the revised main text.

1. In the abstract, we added:

“We experimentally demonstrate in-situ learning capability and achieve competitive classification accuracy (91.71%) on a standard machine learning dataset (MNIST), and confirm the self-adaption of the training algorithm to hardware imperfections.”

2. In paragraph 1 of the introduction (page 3), we added:

“This level of performance is obtained with 11% devices in the crossbar unresponsive to programming pulses and the training algorithm blind to the defectivity, demonstrating the self-adapting capability of the in-situ learning to hardware imperfections.”

3. Revised the following sentence in page 6 to make it clearer that some parts of the calculation are performed in software.

“The hidden neurons after each layer apply a nonlinear activation (in this work, a rectified-linear function in software) to the weighted sums computed in the crossbar. The desired weight update (ΔW) for each layer was calculated in software using Eq. 1, then applied to the crossbar by the measurement system.”

2) The abstract and subsequent descriptions suggest that they are performing “Bayesian Inference.” While in a broad sense of the word this could be said to be the case (most neural networks can be described through a Bayesian perspective), the authors appear to be simply implementing a standard 3 layer neural network (or 2 layers, depending how you count) with a software log-likelihood softmax as the output and input to the training signal. It isn’t immediately obvious that this analysis is strictly Bayesian, aside from their probabilistic normalization in equation 5. At least the way described in the paper, it comes off as standard neural network training with a fancy name. If the authors have a stronger Bayesian rationale or justification, they should be explicit about it.

Response: We agree with the reviewer. In the revised manuscript, we removed the term “Bayesian inference” in order to avoid confusion. Instead, we only use “Bayesian probability” to introduce some calculations (e.g., equation 5).

3) 91.71% accuracy on MNIST is fairly horrible by computer vision standards. Yes, the images were reduced in size to 8x8 to fit on the crossbar; but a generic machine learning researcher will see that number and dismiss the paper. The authors need to acknowledge this discrepancy and address how larger hardware will likely do better.

Response: We thank the reviewer for pointing this out. It reminds us that different communities have different perspectives on MNIST classification accuracy. We understand that the 91.71% accuracy is not on par with a much larger and fully-connected multilayer neural network implemented with mature CMOS technology. However, our work is the first experimental demonstration of emerging-device-based neural network that is much smaller than the CMOS counterparts. The key contribution of this work is the function demo rather than performance optimization of a memristor neural network. We believe that once all key engineering challenges are fully addressed, larger memristor networks that can implement more sophisticated algorithms are feasible. Consequently, a classification accuracy comparable with traditional hardware can be achieved with other benefits such as energy efficiency.

To understand the potential of the memristor array, we developed a simulation model based on measured data (11% unresponsive devices, 2% random variation during blind conductance update, limited conductance dynamic range, etc.) and conducted several simulations. With a 128×64 array (64 inputs, 54 hidden units), the classification accuracy (91.4±1.2%) during the training is very close to experimental data (91.71%). For a much larger network (1024×512 array, 484 inputs, 502 hidden units), our simulation yields an accuracy of 97.3±0.4%. We added the following paragraph in the main text (Pages 10-12) to explain the effects of defectivity, variation and array size:

“To understand the potential of the memristor array, we developed a simulation model (see Supplementary Fig. 7 for detailed architecture) based on measured parameters such as the unresponsive rate, conductance update error, and limited memristor conductance dynamic range. We found that the simulated accuracy agrees well with the experimentally achieved one (Fig. 4a), validating the simulation model. A further simulation on a defect-free network shows that the MNIST classification accuracy is similar to that of the same network architecture trained in TensorFlow⁴⁶ that uses 32-bit floating point numbers. This result suggests that even though the analogue memristor network has limited precision due to conductance variation, they do not have a significant impact on MNIST classification accuracy (more analysis on conductance update variation is shown in Supplementary Fig. 11). Our finding is consistent with previous theoretical and simulation studies on more sophisticated problems^{20,24,47-51}. The analogue precision could potentially be improved, if needed for other applications, by device engineering for better IV linearity^{37,38}, or using pulse width instead of amplitude to represent analogue input (with increased time overhead)^{27,28,37}, or employing multiple memristors to represent one synaptic weight (at the expense of chip area)⁵², etc.

A multilayer neural network trained with online algorithm is more tolerant to hardware imperfections. The experimental accuracy of 91.71% in this work was achieved with 11% devices unresponsive to conductance updates. Our simulation showed that even with 50% of the memristors stuck in a low-conductance state, a 70% classification accuracy is still possible through online training (Fig. 4b), although the accuracy is much more sensitive to shorted devices (Supplementary Fig. 12). On the other hand, if pre-trained weights are loaded to the memristor crossbar (i.e., ex-situ training), the classification accuracy decreases quickly with the

defect rate (Fig. 4b)). There are approaches to improve the robustness of ex-situ training^{19,38}, but most of them require that the parameters be tuned based on specific knowledge of the hardware (e.g. peripheral circuitry) and memristor array (e.g. device defects, wire resistance, etc.), while the in-situ training adapts the weights and compensates them automatically. The self-adaption is more powerful in a deeper network in which the hidden neurons are able to minimize the impact of defects, as suggested by the higher classification accuracy from a two-layer network than that from a single-layer one on the same images (Fig. 4c). The online training is also able to update the weights to compensate for possible hardware and memristor conductance drift over time (see Supplementary Fig. 13).

We expect that the classification accuracy can be improved substantially with a larger network that has more hidden units and/or more inputs to support images with higher resolution. We performed a simulation with our model on a 1024×512 memristor array, which is likely to be available in the near future, to recognize images of 22×22 pixels cropped from the MNIST dataset. The network consists of 484 input neurons, 502 hidden neurons, 10 output neurons, and a total of 495,976 memristors in the two layers to represent the synaptic weights (see Supplementary Fig. 14). After training on 1,200,000 images (20 epochs), a $97.3 \pm 0.4\%$ classification accuracy is achieved on the test set even with 11% stuck devices, approaching that demonstrated with traditional hardware.

”

Fig. 4 Analyses based on experimental-calibrated simulation. **a** The experimental classification error (accuracy shown in Supplementary Fig. 10) matches the simulated accuracy. The simulation considers experiment parameters including 11% devices stuck at $10 \mu S$, 2% conductance update variation, limited conductance dynamic range, etc. The simulation on defect-free assumption shows an accuracy approaching to that from TensorFlow. Each data point is the classification error rate on the complete testing set (10,000 images) after 500 images (simulation or TensorFlow) or 5,000 images (experiment). **b** The impact of non-responsive devices on the inference accuracy with in-situ and ex-situ training approach. The non-responsive device was stuck in a very low conductance state ($10 \mu S$), which is the typical defect device value observed in the experiment. The result shows that the in-situ training process adapts to the defects, providing a much higher defect tolerance compared with pre-loading ex-situ training weights into the network. With 50% stuck OFF devices, the network can still achieve 60% accuracy. **c** The multilayer network also helps with the defect tolerance. If one device is stuck, the associated hidden neuron will adjust the connections accordingly. **d** The simulation of a larger network constructed on a larger memristor crossbar (1024×512) with experimental parameters (e.g. 11% defect rate) could achieve accuracy above 97%, which suggests a large memristor network could narrow the accuracy performance gap from the conventional CMOS hardware. The network architecture is shown in Supplementary Fig. 14.

a. Related, there is a clear precision concern around memristors and similar devices for analog computing of neural networks. There is a growing literature around this subject (hardware papers by Burr and Agarwal and others; machine learning papers by Bengio and others regarding low-precision networks). The authors should include deeper discussions about precision and machine learning.

Response: This is a very good point, and we agree that the precision issue deserves more discussions. Unlike digital neural networks, analog computing in emerging-device-based neural networks is generally of low precision, for both synaptic weight representation and matrix multiplication. As the reviewer pointed out, there is a growing literature on low precision networks. Most agree that low precision networks (and even binary networks) can, under certain conditions, achieve performance comparable to that of a high-precision counterpart.

While CMOS-based accelerators have already started reducing precisions to save chip area and decrease energy consumption, there have been no previous reports of a successful experimental demonstration using emerging devices organized into large arrays. In a pioneering work by Burr et al (cited as ref. 24 in the revised manuscript), researchers built a network using phase change memory (PCM) to store the synaptic weights, with weighted sums performed in software. They discussed the impact of the limited precision of PCM which represents synaptic weight, and concluded the nonlinearity and asymmetry in G-response are the most important factors that limit the maximum possible accuracy to about 85%. Agarwal et al (cited as refs. 50, 51 in the revised manuscript) studied specific requirements for the memristors neural accelerator, including read noise, write noise, asymmetric nonlinearity in the conductance change response to electrical pulses. Our work focusses on the experimental demonstration, and our results agree with their discussions in principle. Specifically we realized a more practical weight update scheme to significantly improve the linearity and symmetry in G-response, and, more importantly, perform the weighted summation (matrix multiplication) experimentally in crossbar. Our experimental result takes into account various factors that diminish precision, including not only the limited precision of weights represented by analog memristors, but also inaccuracies in multiplication due to device IV nonlinearity, wire resistance, thermal noise, etc.

It is also worth noting that, although low precision seems to be sufficient for some networks (e.g. two-layer fully connected layer for MNIST classification), there are approaches to higher

precision if needed. These include engineering memristor devices for a better IV linearity, or using pulse width instead of amplitude to represent analog input (at the expense of increased time and circuitry overhead), or combining several memristors to represent one synaptic weight (at expense of increased chip area), etc. These might be of interest for future work.

In the revised manuscript, we added more discussion as described in *Response to Comment 3*).

4) The energy analysis in both the paper and methods reads a bit like a sales brochure. First, I don't understand why they are comparing the cost of an analog operation to a FLOP, when clearly the analog operation is not at the same precision. Whatever alteration (more memristors?) has to be made to the circuit to achieve comparable digital precision will likely increase cost. There is a literature around this precision consideration that the authors should cite. Further, the conventional CMOS operations are generic, so the extra circuitry that seems to be ignored in the analysis of their own system probably should be accounted for as well (if it is not already, it was not completely clear). I'm also not entirely sure, but is not the number quoted for TrueNorth (>20pJ per operation) for spike events? The whole point of TrueNorth is that it is event driven, thus proportionally a lot more computation occurs with each spike than in a conventional operation.

Again, the value of analog computing is well established if systems can be efficiently trained and scaled using the technology described here. The authors do not need to use poorly set up apples-to-oranges comparisons to make their technology look better than competitors. At least not in an academic publication.

Response: We thank the reviewer for acknowledging the value of memristor based analog computing system, and we respect the reviewer for pointing out the comparisons we make are not entirely fair. We agree that the analog operation has lower precision than a 32-bit FLOP, but in many neural network applications, higher precision could be redundant since it will not necessarily lead to much improved performance. That is why we compared our analog operation with an equivalent operation in digital system (FLOP). Probably for the same reason, TrueNorth, an event-based system, compares its low-precision event-based “synaptic operation (SOP)” with FLOP of conventional architecture supercomputers in the sensational publication^{R1}. However, we do agree that the differences in the system architectures make a fair comparison impossible. We therefore replace the controversial comparisons with some qualitative discussion of the advantages of memristive neural networks in the revised manuscript (Pages 13-14).

“A further potential benefit of utilizing analogue computation in a memristor-based neural network is a substantial improvement in speed-energy efficiency. The advantages mainly come from the fact that the computation is performed in the same location used to store the network data, which minimizes the time and energy cost of accessing the network parameters required by the conventional von-Neumann architecture. The analogue memristor network is also capable of handling analogue data acquired directly from sensors, which further reduces the energy overhead from analogue-to-digital conversion. The memristors we used maintain a highly linear IV relationship, allowing for the use of voltage-amplitude as the analog input for each layer. This also minimizes circuits complexity and hence energy consumption for future hardware hidden neurons and output current readout. While the external control electronics we use in this work is not optimized for fast speed and low power consumption yet, previous

literatures on circuit design^{45,51} and architecture^{53,54} suggest an on-chip integrated system would yield significant advantages in speed-energy efficiency”

We also added the literature on the precision consideration as the reviewer suggested, please see *Response to Comment 3a)* for details.

References

^{R1} Merolla, P. A. *et al.* A million spiking-neuron integrated circuit with a scalable communication network and interface. *Science* **345**, 668-673 (2014).

5) The time analysis suffers from a similar problem. They treat the crossbar as a parallel architecture uniquely, but of course GPUs and CPUs can also be parallelized. The low-level mat-vec kernels are not as easy to split up in conventional treatments, but training batches are routinely distributed over many cores and results then brought back together. It is a different parallel model to be sure, but to say that a GPU can be matched by a 512x256 array ignores the fact that I could theoretically string together many GPUs to parallelize my training over a very large data set, which is not that immediately feasible for the analog system.

Response: We agree that training of neural networks on conventional hardware can be highly optimized/parallelized and it is not reasonable to compare the analog memristor accelerator with a non-optimized assumption. Therefore, we removed the corresponding quantitative comparison in the revised manuscript, as has been discussed in *Response to Comment 4)*.

a. Related, most state-of-the-art neural networks are convolutional (at least in part) or involve more sophisticated micro-networks, like LSTMs. The simplest way to think of these for the analysis here is that these algorithms leverage sparse matrices; which presumably the conventional hardware can take advantage of, while the physical crossbars would benefit minimally from (at least in terms of time).

Response: We agree that our current memristor crossbar structure is most suited to fully connected layers, and this is what we demonstrated in this manuscript. It is certainly possible to implement CNNs and LSTMs on a memristive platform, but the advantages over traditional hardware are likely to be diminished. CNNs, LSTMs, and related systems are useful only for datasets with meaningful spatial organization of the elements of the feature vector, such as image processing and time series analysis. It is possible that CNNs can be implemented with memristors, as have been demonstrated earlier in image convolution with multiple convolutional kernels^{R2,37}. Fully-connected networks, on the other hand, are more useful for the broad class of problems which lack such a spatial structure, and are hence worth studying as well.

References

^{R2} Gao, L., Chen, P.-Y. & Yu, S., Demonstration of Convolution Kernel Operation on Resistive Cross-Point Array. *IEEE Electron Device Letters* **7**, 870-873 (2016)

³⁷ Li, C. *et al.* Analogue signal and image processing with large memristor crossbars. *Nature Electronics* **1**, 52-59 (2018).

All of these points are not to question the overall value of the study. Clearly, the efficient training of analog memristor accelerators for neural networks is a huge and important win. But the authors do not need to oversell this by framing comparative analyses in such a way as to

make best case assumptions for their own technology and worst case for alternatives. It is kind of like putting cheap chrome on a Ferrari. From a distance it may look impressive, but up close it is a waste and distraction.

Response: We thank the reviewer again for pointing out the importance of our work. We hope the revisions we made have satisfactorily addressed the major concerns.

Some minor questions / points:

6) Of the 36 citations, the vast majority are to memristor papers, with 28 in the first three lines of the abstract; whereas there are almost no citations for the algorithm they're using and minimal to the broader machine learning applications (ignoring the growing literature around low-precision ANNs) that could be impacted. For a Nature Communications article, I'd suggest a subset of references suitable for a broad audience, not a narrow community of device researchers. I'd restrict the memristor citations to those that are specifically relevant for the work being described here.

Response: It is a great suggestion to add more citations from machine learning community to address the broader audience of the paper. Specifically, we added the following references, with majority of them from machine learning community. The numbering of the references is updated accordingly in the revised manuscript.

- 7 Krizhevsky, A., Sutskever, I. & Hinton, G. E. in 2012 *Advances in neural information processing systems*. 1097-1105.
- 8 Simonyan, K. & Zisserman, A. Very deep convolutional networks for large-scale image recognition. *arXiv preprint arXiv:1409.1556* (2014).
- 9 He, K., Zhang, X., Ren, S. & Sun, J. in 2016 *Proceedings of the IEEE conference on computer vision and pattern recognition*. 770-778.
- 39 Rumelhart, D. E., Hinton, G. E. & Williams, R. J. Learning representations by back-propagating errors. *nature* **323**, 533 (1986).
- 40 LeCun, Y., Touresky, D., Hinton, G. & Sejnowski, T. in *Proceedings of the 1988 connectionist models summer school*. 21-28 (CMU, Pittsburgh, Pa: Morgan Kaufmann).
- 41 Zhang, Y., Wang, X. & Friedman, E. G. Memristor-Based Circuit Design for Multilayer Neural Networks. *IEEE Transactions on Circuits and Systems I: Regular Papers* **65**, 677-686 (2018).
- 42 Rosenthal, E., Greshnikov, S., Soudry, D. & Kvatinisky, S. in 2016 *IEEE International Symposium on Circuits and Systems (ISCAS)*. 1394-1397.
- 43 Soudry, D., Castro, D. D., Gal, A., Kolodny, A. & Kvatinisky, S. Memristor-Based Multilayer Neural Networks With Online Gradient Descent Training. *IEEE Transactions on Neural Networks and Learning Systems* **26**, 2408-2421 (2015).
- 44 Hasan, R. & Taha, T. M. in 2014 *International Joint Conference on Neural Networks (IJCNN)*. 21-28.
- 45 Gokmen, T., Onen, M. & Haensch, W. Training Deep Convolutional Neural Networks with Resistive Cross-Point Devices. *Frontiers in neuroscience* **11**, 538 (2017).
- 46 Abadi, M. *et al.* in 2016 *Proceedings of the 12th USENIX conference on Operating Systems Design and Implementation*. 265-283 (ACM).
- 47 Courbariaux, M., Bengio, Y. & David, J.-P. Training deep neural networks with low precision multiplications. *arXiv preprint arXiv:1412.7024* (2014).
- 48 Hubara, I., Courbariaux, M., Soudry, D., El-Yaniv, R. & Bengio, Y. Quantized neural networks: Training neural networks with low precision weights and activations. *arXiv preprint arXiv:1609.07061* (2016).
- 49 Gupta, S., Agrawal, A., Gopalakrishnan, K. & Narayanan, P. in 2015 *International Conference on Machine Learning*. 1737-1746.
- 50 Agarwal, S. *et al.* in 2016 *International Joint Conference on Neural Networks (IJCNN)*. 929-938 (IEEE).
- 51 Marinella, M. J. *et al.* Multiscale Co-Design Analysis of Energy, Latency, Area, and Accuracy of a ReRAM Analog Neural Training Accelerator. *arXiv preprint arXiv:1707.09952* (2017).

- 52 Boybat, I. *et al.* Neuromorphic computing with multi-memristive synapses. *arXiv preprint arXiv:1711.06507* (2017).
- 53 Gokmen, T. & Vlasov, Y. Acceleration of Deep Neural Network Training with Resistive Cross-Point Devices: Design Considerations. *Frontiers in Neuroscience* **10** (2016).
- 54 Zidan, M. *et al.* Field-Programmable Crossbar Array (FPCA) for Reconfigurable Computing. *IEEE Transactions on Multi-Scale Computing Systems* **PP**, 1-1 (2017).

7) On the algorithm implementation (Figure 2), how are the errors from each sample accumulated or stored prior to the update weights (once per batch) on hardware? This accumulation is in software, right?

Response: Yes, we used mini-batches for the training on the MNIST dataset. In the present implementation, we calculate the weight gradient using the current delta and x values and accumulate the weight gradient in software (MCU or PC). We are now in the process of designing on-chip integrated peripherals to perform the gradient accumulation in hardware, which we believe is beyond the scope of the present work.

For a smaller dataset, we could get rid of mini-batches, and update the memristor conductance online after each training image. Once we are able to implement sub-microsecond weight updates, minibatches of size 1 will become practical on datasets of size comparable to MNIST. However, in our present peripheral implementation it is more efficient to do inference on multiple images at once, so the mini-batch implementation will speed up the training significantly. To make this point clearer and avoid confusion, we revised the following sentence in Page 6:

“The desired weight update (ΔW) for each layer was calculated in software using Eq. 1, then applied to the crossbar by the measurement system”

8) The ideal algorithm is the exact same network from layer 2A, but performed in floating point in simulation? Or with approximate analog precision minus the defect noise? Did the authors try a state-of-the-art computer vision algorithm (e.g., deep convolutional net) on the lower-resolution images to quantify the impact of the low-res?

Response: The “ideal algorithm” indicates a simulation of an exactly the same network with no defective devices. The inaccuracy of blind conductance updates (which lead to reduced analog precision) and the limitations of memristor dynamic range are both included in this simulation. The purpose of this simulation is only to demonstrate the defect-tolerance of our system. We revised the label in Fig. 3c in the main text and its corresponding figure legend from “ideal algorithm” to “defect-free” simulation.

Following the reviewer’s suggestion, we tried to train the same size of network with *TensorFlow* (32-bit floating point for internal data, no random variation in weight updates, essentially unbounded dynamic range). We found that classification accuracy during the online training from a defect-free simulation is fairly close to training on TensorFlow. We also conducted simulations on higher-resolution images (22×22) with larger networks in a 1024×512 memristor crossbar (484 input neuron, 502 hidden neurons, 10 output neurons, and 495,976 memristors represents synaptic weights). Even with 11% of devices unresponsive and the same limitations imposed on dynamic range and update precision, the simulation delivers 97.3±0.4% accuracy on MNIST.

Please see our *Response to Comment 3* for the simulation results and our corresponding revisions.

9) I'm not sure the confusion matrix (table 1) adds anything here, though it may be interesting if the software-“ideal” case yielded a qualitatively different confusion matrix

Response: Thanks for the comment. We intended to highlight the large amount of experimental data, but we agree it does not tell us anything unexpected, so we have moved the table into *Supplementary Information as Supplementary Table 1*

10) Top of page 10, the authors state “an advantage of online training is the ability to update the weights continuously during use, to compensate for conductance drift over time and changes in the input data distribution”. This is true; however, the authors should point out that neural networks suitable for transfer learning / adapting to concept drift are still very preliminary.

The case they point out about using this online training to compensate for hardware drift is a very important observation, and probably merits more discussion.

Response: We thank the reviewer for the suggestion. In the revised manuscript, we did further simulation to discuss how the training adapts to hardware drift, including hardware peripheral and memristor conductance. We realize an in-depth discussion may be of interest for future work. Therefore, we modified the corresponding sentence in Pages 11-12 of revised main text as follows:

“On the other hand, if pre-trained weights are loaded to the memristor crossbar (i.e., ex-situ training), the classification accuracy decreases quickly with the defect rate (Fig. 4b)). There are approaches to improve the robustness of ex-situ training^{19,38}, but most of them require that the parameters be tuned based on specific knowledge of the hardware (e.g. peripheral circuitry) and memristor array (e.g. device defects, wire resistance, etc.), while the in-situ training adapts the weights and compensates them automatically. ”

“The online training is also able to update the weights to compensate for possible hardware and memristor conductance drift over time (see Supplementary Fig. 13)”

Supplementary Figure 13. Simulated self-adaption to hardware and memristor conductance drift. *a*, The in-situ training of the memristor network adapts a simulated change in current sensing. We changed a different scaling factor with mean of 1.0 and s.d. of 0.3 on each dimension of the output vector, to simulate a possible hardware drift of the transimpedance amplifier (TIA). The 100-repeated simulation result shows the training algorithm adapts the peripheral asymmetry after training on several samples. The cyan dashed line shows the accuracy before the hardware drift. *b*, The in-situ training of the network adapts the memristor conductance drift. The simulated array was constructed using the same parameters from the experiments, and before the conductance drift the network accuracy is indicated by the cyan dashed line. After the conductance changes by the factor with mean of 0.8 (red)/1.1 (blue) and standard deviation (s.d.) of 0.2 (to simulate a pretty bad case), the in-situ training of the network quickly adapts after training a few samples (plot shows 100 repeated simulation). It is noteworthy that the amount of the simulated conductance drift is unlikely in an experimental array, but we simulated a worse case to make the self-adaption process more noticeable.

11) Page 11 (And in methods) As mentioned above, TrueNorth’s power consumption is ~22pJ per spike event, not per algorithmic operation (the event-driven nature of the chip means that they pay ~0pJ when nothing is communicated). This is a subtle but very important distinction – it shifts the burden of energy use onto the algorithm; but it is not fair to imply that TrueNorth is 1000 times more power hungry per operation.

Response: We thank the reviewer for the comments. We agree that TrueNorth is an event-based system, but they themselves compared their low-precision event-base “synaptic operation (SOP)” with FLOP of conventional architecture supercomputers in their publication¹. Nevertheless, we agree that our power consumption is not complete given we could not include an accurate estimation of power consumption in the peripheral circuits yet. Therefore, we replaced the corresponding comparison with qualitative discussion in the revised manuscript (Pages 13-14).

Reference:

1 Merolla, P. A. *et al.* A million spiking-neuron integrated circuit with a scalable communication network and interface. *Science* **345**, 668-673 (2014).

12) It is probably more fair to communicate “error rate” on MNIST as opposed to “accuracy”

Response: Thank the reviewer for the suggestion. We believe the accuracy is a more straightforward measure for a broad audience, but we agree that the error rate delivers more technical information. In the further analysis we added in the revised manuscript, we used the error rate in a log-log plot in Fig. 4a in the revised manuscript to compare the experiment data with various simulations and software.

Reviewer #2 (Remarks to the Author):

1. The paper presents a memristive neural network with linear relation between the voltage and the memristor conductance and therefore it can be trained in-situ reliably.

Overall, the paper is well-written and the results are interesting and important. The reviewer thinks that this work is with high quality and deserves publication.

Response: We thank the reviewer for the positive comments and recommendation.

2. The reviewer feels that there is too much emphasize on the neural network theory rather than focusing on the contributions of the paper. The authors should focus more about the fabrication and process of the memristor array and PCB in use since the design and structure are not clearly described.

Response: We thank the reviewer for the useful suggestions. We added the detailed fabrication process as *Method 1* in the revised main text. We also added the following schematic for the design of measurement system to the SI as *Supplementary Fig. 4*.

“1. Device fabrication and array integration

The transistor array and interconnection between cells are taped out from a commercial foundry with 2 μm technology node to achieve low wire resistance. We monolithically integrate memristors on top of as-received chip in house. Pd/Ag contact metals are firstly deposited on both vias after argon plasma treatment to remove the native oxide. The chip is then annealed at 300 °C for 30 min in 20 s.c.c.m nitrogen flow to achieve good electrical contact. The memristor bottom electrode is deposited by evaporating 20 nm Pt on top of a 2 nm thick Ta adhesive layer and patterned by photolithography and lift-off in acetone. A switching layer of 5 nm thick HfO₂ is deposited by atomic layer deposition using water and tetrakis(dimethylamido)hafnium as precursors at 250 °C, and then patterned by photolithography and reactive ion etch (RIE). Finally, the top electrode of 50 nm thick Ta is deposited by sputtering and liftoff, followed by sputtering of a 10 nm thick Pd protection layer.”

Supplementary Figure 4. Circuit diagram of the measurement system. MATLAB script with graphic user interface (GUI) running on a general-purpose computer is used to communicate with the microcontroller through a serial port for various applications. The microcontroller controls all the circuit components on the measurement system, such as DACs (Digital-Analog Converters) for generating driving pulses and ADCs (Analog-Digital Converters) for collecting the measurement data. Note that voltages applied on either rows or columns on the memristors crossbar, or gates on the transistors are generated in parallel by different DACs that located on the rows and columns boards. Four transimpedance amplifiers (TIAs) with different gains are attached to each column to cover 4 orders of magnitude current ranges, but only one TIA is chosen during the operation depending on the output current level. During a read or vector-matrix multiplication (VMM) operation, small voltages (representing input vectors) are applied on the corresponding rows (marked as BEs) by the DACs. Currents (representing output vectors) are read out by the TIAs and ADCs on columns (marked as TEs). The readout currents are processed in the microcontroller, and/or sent back to the general-purpose computer. The write operation is similar to the reading. Positive voltages can be applied on selected rows for reset or selected columns for set, while all other unselected terminals can be grounded, floated or biased with an arbitrary voltage value.

A schematic of the 1T1R arrays should be provided, including showing how the same memristive array is used for different structures of neural networks (how a 128x64 array becomes 64x54x10 two layer MNN?). Fig. 2 shows pure crossbars that are very different than the real architecture used in this work. Fig. 5 in supp. also seems to present only part of the story. **Response:** We provide an additional schematic of the 1T1R array as *Supplementary Fig. 4* (see previous response above). We revised the main text in Page 9 and Fig. 3b to make it clearer how a 128x64 array was partitioned for 64-54-10 two-layer network:

“The two-layer network used 7,992 memristors (see Fig. 3b for the partition on a 128x64 array), each of which was initialized with a single pulse with a 1.0 V gate voltage from a low conductance state.”

We also added schematic to show how to perform inference (matrix multiplication) in the crossbar, as new panel **a** in the revised *Supplementary Fig. 5*.

Supplementary Figure 5. Illustration of proposed ITIR operation by electrical pulses. **a**, The matrix multiplication is performed by applying the input voltages (the amplitude of them represents the vector values) on the row wires, and reading the current on the column wires representing the output current. All the transistors are turned on by applying synchronized pulses on the word-line wires, making the ITIR array a pseudo-crossbar. **b**, Parallel-reset scheme. A reset pulse is applied to the bottom electrodes of an entire column simultaneously, while the top-electrode voltages are used to control which devices are reset. **c**, Parallel-set scheme. Pulses are applied to the top electrodes of an entire row simultaneously, while the gate voltages vary along the row.

Also the linear programming approach should be better explained. Specifically, how many levels can be achieved? (from Fig. 1, ~100 levels are shown).

Response: We thank the reviewer for suggesting that we further explain of the weight update mechanism. Linear programming is critical to the successful demonstration of memristor neural network. In the main text (Page 4), we have the following sentences to explain how the linear programming approach works:

“To increase the conductance of a given cross point, we applied synchronized positive voltage pulses from a driving circuit board to the memristor top electrode and the gate of the series transistor. The gate voltage, which specifies a compliance current, determines the resulting memristor conductance. We decreased the conductance by first applying a sufficient positive pulse to the memristor bottom electrode to initialize the state, and then used the conductance increase scheme to set the memristor to the desired level (illustrated in Supplementary Fig. 1). With this scheme, we achieved linear and symmetric conductance increase and decrease with minimal cycle-to-cycle (Figs. 1f, 1g) and device-to-device (Figs. 1f, 1h) variations.”

We also would like to point out that it is an analog system, so in practice we do not use discrete conductance levels. Also, because during training each weight is adjusted many times, much higher effective precision can be achieved. The ultimate limitation of the precision of the memristor-based synapse is determined by the stability of the device conductance. Our previous study on the same device suggests a 7 bit/128 levels precision, as shown in the following figure. This value could be potentially further improved with novel devices / materials.

Fig. RI | g, Room-temperature state retention and read disturb of the device states. The d.c. conductance states of all devices were measured with a 0.2 V bias for 1,000 cycles, or a total of 6.4 h, showing no discernible drift in the plots. **h**, Histogram of the normalized standard deviation (s.d.), defined as the s.d. per conductance range (100–900 μS), for all measured states, which was fitted to a lognormal distribution. This shows that there are fluctuations during the read operation that can occasionally degrade the effective precision of an individual memristor, but 90% of the device states have a normalized s.d. less than 0.39%. (Taken from ref. 37)

Reference:

37 Li, C. *et al.* Analogue signal and image processing with large memristor crossbars. *Nature Electronics* 1, 52-59 (2018).

3. It will be worthwhile to provide the code for the software implementation of the training and the missing circuit units.

Response: Thank the reviewer for the comment. We added the following code snippet in the revised *Supplementary Note 1*

Main loop of the training algorithm:

```
% The main training loop:
for j = 1:n_batches % For each batch

    % Perform inference in the crossbar
    batch = data(:,(j-1)*batch_size+1:j*batch_size);
    [outs, voltages] = obj.test(batch);

    grad = z; % Initialize cumulative gradient to all zeros

    for k=1:batch_size
        %For each example, compute the gradient estimate by backpropagation in software:
        I_out = outs(:,k);
        indx = (j-1)*batch_size+k; % Compute index of this training example

        % Get voltages for this example
        V = cellfun(@(x) x(:,k), voltages, 'UniformOutput', false);
        % Get gradient based on this example
        temp = obj.calculate_update(V, I_out, Labels(indx), w);
        % Add it to the cumulative batch gradient
        grad = cellfun(@plus, grad, temp, 'UniformOutput', false);
    end

    % Apply the weight update in crossbar:
    obj.update_weights(grad, rate/batch_size, j);
    [w, rawG] = obj.get_weights('mode','fast'); % Read new weights
end
```

Forward pass:

```
for i=1:obj.n_Layers % For each Layer
    if any(abs(voltages{i}(:)) > obj.V_read)
        % Clipping signals to protect the array operation
        voltages{i}(voltages{i} > obj.V_read) = obj.V_read;
    end

    [currents{i},raw_out{i}] = obj.array.subs{i}.read_current([voltages{i}; -voltages{i}]);
    voltages{i+1} = obj.activation(currents{i});
end
I_out = currents{end};
```

The software implementation of ReLU and its derivative:

```
c = 200; % The scaling factor that convert a current to a voltage
activation = @(x) c*max(0, x); % ReLU
aderiv = @(x) c*(x>0); % derivative of ReLU
```

Backward pass:

```
p = zeros(1, length(I_out));
for i=1:length(I_out)
    p(i) = 1./sum(exp(k*(I_out-I_out(i)))); % Softmax
end

deltas{end} = -p;
deltas{end}(Label) = (1-p(Label));

for i=numel(deltas):-1:1
    grad{i} = (voltages{i}*deltas{i})'; % Transpose to line up with weights
    if i>1
        deltas{i-1} = ((deltas{i} * weights{i}) .* obj.aderiv(voltages{i}'));
    end
end
```

4. The presented learning scheme with fixed pulses across the memristor and with varying pulse magnitude on the gate of the transistor is similar to a stochastic gradient descent training technique presented in Soudry et al., "Memristor-based Multilayer Neural Networks with Online Gradient Descent Training," TNNLS 2015. It worth mentioning the differences and similarities. They also provided the circuits for the training in Rosenthal et al., "A Fully Analog Memristor-Based Multilayer Neural Network with Online Backpropagation Training", ISCAS 2016.

Response: We thank the reviewer for providing these two important papers. In the revised manuscript, we cite the two papers (Soudry et al, TNNLS 2015 and Rosenthal et al, ISCAS 2016) as ref. 42 and ref. 43 in the revised manuscript. The two papers report a complete circuit design for training a fully analog memristor based multiplayer neural network. In their work, a memristor that connected to two transistors (one n-channel and the other p-channel) and use duration of the applied programming pulse to enable the analog conductance change. We, on the other hand, employ one-transistor one-memristor synapse cells, and gate voltage amplitude to control the conductance. In addition to the differences in specific circuit design, the two reports provides feasible approaches to train a multilayer neural network, with emphasis on algorithm development and circuit design. While our current work, on the other hand, focuses more on the experimental implementation.

In the revised manuscript, we added one sentence in the end of the following passage in Page 7.

"The error backpropagation^{39,40} in this work is calculated in software from the values of the readout weights (see Method: 6). In the future, backpropagation can be implemented within the memristor crossbar by applying a voltage vector representing the current-layer error to the

bottom electrodes of the crossbar and reading out the current vector from the top electrodes for the previous-layer error. An on-chip integrated peripheral for full hardware implemented functionality is under development, which has been discussed and simulated in literatures as well⁴¹⁻⁴⁵. ”

5. While some of the results are trivial (the influence of stuck at ON is more significant than stuck at OFF for example is intuitive), others are interesting. The authors should emphasize what are the important insights and what is only a demonstration of known insights.

Response: We thank the reviewer for the constructive suggestion to improve the quality of the manuscript. We rewrote the discussion section in the main text to make the manuscript more focused on the important insights, and here is a summary:

Important insights:

1. Experimentally demonstrated in-situ training of a multilayer memristor neural network can achieve good accuracy, even in the presence of various non-ideal factors, e.g. limited precision of memristor conductance and matrix multiplication, limited dynamic range of memristor conductance, wire resistance, and asymmetric peripheral circuit, etc.
2. An experimentally-validated simulation suggests a larger network in the near future will close much of the gap with conventional CMOS hardware.

Demonstration of known insights with our experimental-validated simulation:

1. The training algorithm is tolerant of defects including the stuck devices, and the influence of stuck ON device is more significant than the stuck OFF devices; The comparison between the influence of stuck ON and stuck OFF devices is moved to the Supplementary Information;
2. The multilayer network is better able to compensate for defects than the single layer network;
3. Small weight update noise helps the training converge, while large noise has an adverse impact. We moved the discussion on weight update noise to the *Supplementary Information as Supplementary Figure 11*.

6. The comparison to ex-situ can be explained better. Also, note that there are work arounds that can improve the vulnerability of ex-situ training to the memristive system faults.

Response: Thanks for the reviewer for reminding us there are work arounds that improve the vulnerability of ex-situ training. We were aware of them and discussed our approach in previous work on ex-situ training, but it requires the knowledge of specific hardware and memristor array. We agree it is a good point worth further discussion, and thus we added more details about the ex-situ discussion in the revised main text on Pages 11-12, as shown below:

“On the other hand, if pre-trained weights are loaded to the memristor crossbar (i.e., ex-situ training), the classification accuracy decreases quickly with the defect rate (Fig. 4b)). There are approaches to improve the robustness of ex-situ training^{19,38}, but most of them require that the parameters be tuned based on specific knowledge of the hardware (e.g. peripheral circuitry) and memristor array (e.g. device defects, wire resistance, etc.), while the in-situ training adapts the weights and compensates them automatically. ”

7. Method 6- “With integrated hardware neurons, we estimate that the time needed for each layer of neurons to stabilize can be brought as low as 10 ns” need to justify this claim.

Response: We thank the reviewer for the comment. We measured the speed of the operation in the presence of all the parasitic on chip, as shown in the *Supplementary Fig. 16* of ref. 37 (shown as Figure R2 here). The intrinsic read operation time of the chip is faster than the 125 ns settling time of the equipment. We expect that with integrated driving peripheral circuits built with a more advanced technology node the readout time would be much faster. We understand designing fast on-chip integrated CMOS circuitry will involve a significant amount of work, which is beyond the scope of this work. Therefore, we removed the comparison based on the 10 ns read time in the revised manuscript.

Figure R2. Speed of the read operation of memristor in the 128×64 array. **a**, The read speed of the memristor in the far end corner of a 128×64 array when the transistor is on (this is what limits the VMM operation speed). The onset oscillations within 100 ns is not caused by the chip, but rather the measurement setup, because the settling time of the equipment (Keysight B1530), according to its specification, is 125 ns. (Adapted from ref. 37)

The revised discussion paragraph on pages 11-12 is as follows:

“A further potential benefit of utilizing analogue computation in a memristor-based neural network is a substantial improvement in speed-energy efficiency. The advantages mainly come from the fact that the computation is performed in the same location used to store the network data, which minimizes the time and energy cost of accessing the network parameters required by the conventional von-Neumann architecture. The analogue memristor network is also capable of handling analogue data acquired directly from sensors, which further reduces the energy overhead from analogue-to-digital conversion. The memristors we used maintain a highly linear IV relationship, allowing for the use of voltage-amplitude as the analog input for each layer. This also minimizes circuits complexity and hence energy consumption for future hardware hidden neurons and output current readout. While the external control electronics we use in this work is not optimized for fast speed and low power consumption yet, previous literatures on circuit design^{45,51} and architecture^{53,54} suggest an on-chip integrated system would yield significant advantages in speed-energy efficiency”

Reference

37 Li, C. *et al.* Analogue signal and image processing with large memristor crossbars. *Nature Electronics* 1, 52-59 (2018).

8. Method 6+7- comparison to execution time of TPU and GPU, increasing the size of the layers till you meet the point where the suggested work outperform GPU\TPU is “artificial”, consider checking the performance for well-known networks such as AlexNet, VGG, ResNet (even only-compare execution of “partial” net).

Response: We thank the reviewer for the suggestions. We agree that the performance comparison is preliminary. And in order to construct very large networks such as AlexNet, VGG, ResNet, etc., we will probably need a complete design which includes significantly larger arrays that communicate with each other. However, it is beyond the capability of our current setup. We therefore removed all the quantitative comparison in the revised manuscript.

We believe that once all key engineering challenges described in this work are fully addressed, it will be straightforward to scale the technologies we develop to larger networks and more sophisticated algorithms to achieve a classification accuracy comparable with traditional hardware with other benefits such as energy efficiency. We added the references to AlexNet, VGG, and ResNet as refs. 7-9 in the revised manuscript.

9. The energy comparison is not convincing. It seems that the authors ignore the energy of the periphery including costly ADCs and voltage/current sources. A real comparison would be comparing the energy per image, but it requires the entire implementations.

Response: We thank the reviewer for the suggestions. It is a legitimate concern that various peripherals such as A/D, D/A converters, voltage/current sources consume a major portion of the power used in a mixed digital-analog signal system. We are working on the design of on-chip integrated peripherals, and once these designs are complete we will be able to test power-consumption predictions experimentally. We also note that in principle, an analog memristive neural network could handle analog data from sensors directly, as we stated on Page 13–14. This full analog system does not need the A/D and D/A circuitry at all. In addition, although beyond the scope of this work, the future peripheral circuits are not limited to CMOS technology only. For example, there were reports that use memristor as neurons to facilitate inference/training^{R1,R2} for a fully analog network, and to construct memristor-based A/D and D/A for mixed-signal networks^{R3}. On the other hand, we agree with the reviewer that the energy comparison without the peripheral circuits is not fair, we have taken this part out of the revised manuscript.

References:

- R1 Tuma, T. *et al*, “Stochastic phase-change neurons”, *Nature Nanotechnology*, 11, 693–699 (2016)
- R2 Wang, Z. *et al*, “Fully memristive neural networks for pattern classification with unsupervised learning”, *Nature Electronics*, 1, 137–145 (2018)
- R3 Gao, L. *et al*, “Digital-to-analog and analog-to-digital conversion with metal oxide memristors for ultra-low power computing”, NANOARCH 2013

Reviewer #3 (Remarks to the Author):

The paper “Efficient and self-adaptive in-situ learning in multilayer memristor neural networks” by Can Li et al. shows a backpropagation implementation on software-hardware implementation. Although interesting, the paper does not show enough novelty to be published in Nature Communications.

Response: We appreciate the reviewer’s time in reviewing the manuscript. In this work, we implemented backpropagation in emerging device hardware with some control peripherals communicating with a general-purpose computer. The significance of memristor-based analog in-memory computing is well established in previous literature, and is recapped here as follows. Metal-oxide memristors and charge-based transistors work under fundamentally different mechanisms. The former is a promising beyond-CMOS device with excellent scalability and 3-dimensional stackability that make it possible to store the massive number of deep neural network parameters in dense arrays. The tunable resistance states can be used not only to store information, but also to perform computation at the same location, circumventing the so called ‘von-Neumann bottleneck’. Matrix multiplication in a crossbar is performed in the analog domain following physical laws, such as Ohm’s law for multiplication and Kirchhoff’s current law for summation in a parallel fashion. The analog computing capability of the memristor allows it to handle analog signals directly without digital conversion and suggests further increase in speed-energy efficiency.

There is no doubt the concept of memristor-based analog in-memory computing is still in early stage of development. Implementing a fairly large multilayer memristor neural network for computing is hence a significant step forward, as commented by the first two reviewers. Specifically, our contribution in this manuscript include:

1. We demonstrated a linear and symmetric programming of memristor conductance, which is critical to simplifying training of a neural network with general algorithms.
2. We experimentally demonstrated a multiple fully-connected layers in a 128x64 array with in-situ learning capability, while previous implementations are limited to small arrays and single layers.
3. For the first time, we demonstrated the emerging-device-based realization of a modern machine learning algorithm (SGD) for a large standard dataset (MNIST).

1- Please clearly state in the main paper the device programming speed (500us and 5us, as I believe is described in the Supplementary Info), which definitely seems very long for promising very fast training speed.

Response: Thank the reviewer for the comments. We understand that the programming pulse width is long for the current work. This is limited by the current version of our off-chip PCB board-based measurement system, which is designed for great flexibility and large dynamic range often needed in demonstrating different computing functions but not optimized for performance yet. The programming speed of individual devices connected with a series transistor on the chip can be programmed with pulse with below 100 ns (see *Supplementary Fig. 16* in *ref. 37*), and the same discrete device can be programmed within 5 ns (see *ref. 36*).

We are well aware of the speed limitations of the current measurement system and are currently designing and developing application-specific on-chip integrated peripherals that will deliver much faster programming pulses. The focus of the current work is to experimentally

demonstrate the feasibility of an emerging device based analog neural network; performance optimization is a goal for future work.

Figure R2. Speed of the read operation of memristor in the 128×64 array. a, The read speed of the memristor in the far end corner of a 128×64 array when the transistor is on (this is what limits the VMM operation speed). The onset oscillations within 100 ns is not caused by the chip, but rather the measurement setup, because the settling time of the equipment (Keysight B1530), according to its specification, is 125 ns. (Adapted from ref. 37)

Figure R3 (c) The device can be repeatedly switched between HRS and LRS with 5ns pulses (SET: 2.2V; RESET: -4V) indicating faster than 5ns switching speed. (Adapted from ref. 36)

Reference

- 36 Jiang, H. *et al.* Sub-10 nm Ta Channel Responsible for Superior Performance of a HfO₂ Memristor. *Sci Rep* **6**, 28525 (2016).
- 37 Li, C. *et al.* Analogue signal and image processing with large memristor crossbars. *Nature Electronics* **1**, 52-59 (2018).

In order to make the point clearer, we made revisions as follows in Page 14:

“While the external control electronics we use in this work is not optimized for fast speed and low power consumption yet, previous literatures on circuit design^{45,51} and architecture^{53,54} suggest an on-chip integrated system would yield significant advantages in speed-energy efficiency”

2- In the entire paper, many crucial aspects are explained by saying that there will be efficient circuitry able to perform such aspects. Authors should explain how they actually think to implement such aspects, if A/D D/A converters are used, if operation is performed in parallel or serially, since using such high-density arrays pose severe silicon area constraints, and therefore implementable CMOS functionalities.

Response: We thank the reviewer for the suggestions. It is a legitimate concern that a parallel A/D, D/A converter implementation will be too expensive in terms of power consumption and chip area for a mixed digital-analog signal system. Actually there are a number of viable solutions proposed in previous publications^{R1} (e.g. reduce bit precision, share some A/D to make pseudo reduce the degree of parallelization, etc.) to address this issue. We are working on the design of on-chip integrated peripherals, and once these designs are complete we will be able to test power-consumption predictions experimentally. We also note that in principle, an analog memristive neural network could handle analog data from sensors directly, as we stated in Page 13-14. This full analog system would not need the A/D and D/A circuitry at all. In addition, although beyond the scope of this work, the future peripheral circuits are not limited to CMOS technology only. For example, there were reports that use memristor as neurons to facilitate inference/training^{R2,R3} for a fully analog network, and to construct memristor-based A/D and D/A for mixed-signal networks^{R4}. On the other hand, we agree with the reviewer that the energy comparison without the peripheral circuits is not fair, we have taken this part out of the revised manuscript.

References:

- R1 Gokmen, T. et al, "Training Deep Convolutional Neural Networks with Resistive Cross-Point Devices", *Front. Neurosci.*, 11, 538 (2017)
- R2 Tuma, T. et al, "Stochastic phase-change neurons", *Nature Nanotechnology*, 11, 693–699 (2016)
- R3 Wang, Z. et al, "Fully memristive neural networks for pattern classification with unsupervised learning", *Nature Electronics*, 1, 137–145 (2018)
- R4 Gao, L. et al, "Digital-to-analog and analog-to-digital conversion with metal oxide memristors for ultra-low power computing", *NANOARCH* 2013

3- The training is performed using mini-batches. How can the different delta and x values be memorized, before applying the weight update? Also Fig. 2c does not mention this. Where these values will be memorized in circuitry?

Response: Yes, we used mini-batches for the training on the MNIST dataset. The present implementation is to calculate the weight gradient with the current delta and x values and accumulate the weight gradient in software (MCU or PC). The on-chip integrated peripherals we are currently working on would be able to perform the gradient accumulation in hardware.

For a smaller dataset, we could get rid of mini-batches, and update the memristor conductance online after each training image. Once we are able to implement sub-microsecond weight updates, minibatches of size 1 will become practical on datasets of size comparable to MNIST. However, in our present peripheral implementation it is more efficient to do inference on multiple images at once, so the mini-batch implementation will speed up the training significantly. To make this point clearer and avoid confusion, we revised the following sentence on Page 6:

“The desired weight update (ΔW) for each layer was calculated in software using Eq. 1, then applied to the crossbar by the measurement system”

4- Weight update is performed row by row in the scheme in Fig. S5, which slows down programming operation by orders of magnitude (considering the worst case of 500us per line with a 1024x1024 array, this means 500ms, definitely not useful). In addition, Fig. S1 shows that different conductance levels are obtained by programming different gate voltages, which means that the circuit overhead is unacceptably large.

Response: We agree that weight update with 500 us pulse in a final product would be unacceptable. But as we explained previously, the long programming time is limited by the off-chip measurement system that is not optimized for speed. The devices/arrays are capable of much faster operation. It is also noteworthy that we used analog memristor conductance, so each memristor could store 5-6 bit information as suggested in our previous study, which means one conductance update is equivalent to toggling multiple digital cells.

We also agree that the programming method we proposed in this work will increase the circuit overhead in a generic design. An application-specific design to minimize the circuit overhead will be of interest for the future work. We want to mention again that the future peripheral circuits are not limited to CMOS technology only, as we stated in the response to comment 3.

To make the claim clearer and avoid confusion, we have revised the manuscript as described in *Reponses to Comment 1* and *Comment 3*.

5- I don't understand how the programming scheme works: if I program a device into a particular conductance, and then, i.e., I ask to reduce it, do I need to read the device conductance between the two operations to calculate the exact voltage pulse to be applied?

Response: No, we do not read the device conductance during the training, i.e. what we proposed is different from a write-and-verify scheme. During the online training, the algorithm only instructs the hardware of the direction and the amount of the conductance change. The conductance update scheme is described in Method 5 (specifically in Eq. 8). So, if a device conductance needs to be increased, we will add the calculated gate voltage change (ΔV_{gate}) to the last applied gate voltage, and apply the synchronized gate voltage and write/reset voltage to the crossbar.

6- Regarding the obtained accuracy of 91.71% (which actually shows values fluctuating below 90% in Fig. 3c), how it is compared with an equivalent size software network? Because the Ideal Software simulation in Fig. 3c is misleading, since it corresponds to a memristor network with no dead or stuck-on devices, not a software (i.e. Tensorflow) simulation.

Response: We thank the reviewer's suggestion to make comparison with software. First, the accuracy shown in Fig. 3c is the smoothed (*rlowess* methods using span of 10% of the total number of data points) mini-batch accuracy, and we have the test accuracy after training on every 5,000 images in the *Supplementary Fig. 9*, which clearly shows the accuracy is above 91% after 60k training images.

Supplementary Figure 9. Classification test on the entire separate testing set after training on every 5,000 samples. The testing accuracy increases as the number of training samples increases, saturating at around 91%.

As suggested by the reviewer, we changed the legend of “ideal software” in the Fig. 3c to “defect-free simulation”. We further extend our experimentally-validated simulation to higher-resolution images (22×22) and larger network in a 1024×512 memristor crossbar (484 input neuron, 502 hidden neurons, 10 output neurons, and 495,976 memristors represents synaptic weights). With device behavior parameters matched to our experiment, the simulation delivers 97.3±0.4% accuracy on MNSIT.

We revised the manuscript with the new simulation data, as shown on pages 11-12 in the main text.

7- The conductance values shown are very large, hundreds or thousands of uS. This leads to a huge power consumption during programming operation. Did authors consider this in their power estimates?

Response: Our original quantitative power estimation is based on the energy used per inference operation. Those numbers are based on the measured conductance values of the memristors after training. We did not attempt to calculate the power consumption during training, since the power consumed by the programming operation is heavily dependent on the switching speed. Since the memristor we used has demonstrated fast switching, the power consumption during the programming will be significantly reduced with an on-chip integrated peripheral optimized for fast-speed. Nevertheless, we have removed our quantitative comparison in the revised version of main text to avoid confusion. The revised discussion (in Page 13-14 of the revised manuscript) is shown as follows:

“A further potential benefit of utilizing analogue computation in a memristor-based neural network is a substantial improvement in speed-energy efficiency. The advantages mainly come from the fact that the computation is performed in the same location used to store the network data, which minimizes the time and energy cost of accessing the network parameters required by the conventional von-Neumann architecture. The analogue memristor network is also capable of handling analogue data acquired directly from sensors, which further reduces the energy overhead from analogue-to-digital conversion. The memristors we used maintain a highly linear IV relationship, allowing for the use of voltage-amplitude as the analog input for

each layer. This also minimizes circuits complexity and hence energy consumption for future hardware hidden neurons and output current readout. While the external control electronics we use in this work is not optimized for fast speed and low power consumption yet, previous literatures on circuit design^{45,51} and architecture^{53,54} suggest an on-chip integrated system would yield significant advantages in speed-energy efficiency”

8- Please clearly state that Fig. 4 are simulations. What is the real random variation during programming operation of real memristors used in this paper? I think is much more than the optimum 1%.

Response: We thank the reviewer’s comment. To avoid confusion, if any, we added legend of “simulation” to the original figure. In the text, we have already stated the analysis is through simulation.

The random variation during programming of the memristors used in this paper, as analyzed from the data presented in Fig. 1f of the main text, is plotted in the following figure. The s.d. of weight update variation over the conductance range is mainly within 1-3%, depending on the targeted conductance (a higher conductance range yields a narrow distribution). However, the experimentally achieved results agrees well with the simulation using a 2% variation. This result suggests that MNIST classification accuracy is not as sensitive to the real variation as we thought. Future work with optimized devices and systems with well-defined variations, is needed to calibrate the experimental and simulation results.

Figure R4. Random variation during the blind weight update. The analysis is conducted on the data shown in Fig. 1f (main text) for all the devices in the array.

9- Calculations for power consumption completely neglect CMOS circuitry, which can be a large error in case A/D D/A converters are used. Calculations should be redone considering circuitry. In addition, here memristor power consumption is not based on the memristors of the papers, but on the best obtained in the literature. What is the linearity of this cited device, which is the main advantage of the present manuscript memristor?

Response: As we pointed out in the *Response to Comment 2*, the analog memristive neural network could handle analog data directly without digital conversion, and the peripheral could be based on emerging device and not limited to CMOS technology. The contribution of this work is to experimentally demonstrate the feasibility of core computation in the memristor crossbar. It

includes performing inference (matrix multiplication) in a massive parallel way and updating weights that self-adapts the various kind of defects.

The estimated energy cost for weight update in the original manuscript was based on the most energy-efficient memristor with good linearity from literature (ref. 34), although its analog switching potential is unclear yet. The power consumption for inference operation was based on our own measured data. Nonetheless, we agree with the reviewer that without a complete peripheral circuitry the comparison is not totally fair. Therefore, we have removed the quantitative comparison in the revised manuscript.

References

- 27 Yao, P. *et al.* Face classification using electronic synapses. *Nat Commun* **8**, 15199 (2017).
- 28 Sheridan, P. M. *et al.* Sparse coding with memristor networks. *Nature Nanotechnology* **12**, 784-789 (2017).
- 34 Choi, B. J. *et al.* High-Speed and Low-Energy Nitride Memristors. *Advanced Functional Materials* **26**, 5290-5296 (2016).

10- How is the 20ns estimate calculated for read time for CMOS circuitry?

Response: Thank the reviewer for the comment. We measured the speed of the operation in the presence of all the parasitic on chip, as shown in the Figure R1 in *Response to Comment 1*). The intrinsic read operation time of the chip is faster than the 125 ns settling time of the equipment. We estimate with integrated driving peripheral circuits in a more advanced technology node the readout time would be much faster. Nonetheless, we removed the calculation based on the 20 ns read time in the revised manuscript to avoid confusion.

REVIEWERS' COMMENTS:

Reviewer #1 (Remarks to the Author):

I think this manuscript is improved from the last version and I appreciate the authors responsiveness to my concerns about the the neural network approach and the apples-to-apples comparison to alternative technologies.

The paper is well-suited for publication in my opinion. My one request would be that the authors include their answer to my comment 5a in the main text of the manuscript. Briefly, the authors look here at neural networks with fully-connected layers, and thus their potential benefit to convolutional and locally recurrent networks (such as LSTMs) are less obvious. That is fine for the state of this research and the finding (I can imagine a cleverly shaped memristor xbar configuration nicely suited for convolutions, for instance); however it is important that the general reader recognizes the difference. Most state-of-the-art neural network applications use these more sophisticated architectures; only a small number of the final layers in large networks like VGG or AlexNet are typically fully connected.

In addition, I think it may behoove the authors to mention that many of the algorithmic design choices for ANNs are determined in part by what is efficient on hardware. And if crossbars are fundamentally different than architectures that benefit from sparse matrix multiplications, that may change what configurations make sense (say, larger convolutional kernels?)

This does not diminish the impact of the paper, but does communicate its potential impact more appropriately to the broader readership of this journal.

Reviewer #2 (Remarks to the Author):

The authors have answered all of my concerns and made a comprehensive revision.

Reviewer #3 (Remarks to the Author):

- 1- I understand the authors' point of view regarding device programming speed, however they show 10 cycles at 5 ns, while the device would undergo millions of cycles during Neural Network Training, with endurance issues not clear yet.
- 2- Authors state that x and Δ values are stored in software. I understand that this approach is easier for implementing the actual experiment, however it is not practical with the actual chip, since it would unacceptably slow down the overall operation. How the authors in practice think to implement this operation in chip? Saving the values in capacitors, with the risk of leakage, or using ADCs to store x and Δ in a near-located digital memory?
- 3- Programming scheme seems very slow (one row each time step). I fear this can be a conceptual problem, since this slows down the speed of training. My question to authors is: what is the possible commercial purpose of this memristor chip? To compete with GPUs or to focus on different applications? Authors should clearly state this, since they claim for "high speed-energy efficiency". I fear is not simple to overcome GPUs with row by row programming.
- 4- Fig. 4a is very useful, thank you for providing it. Experimental accuracy is around 5% below the corresponding TensorFlow accuracy, which seems a large gap, however.

Based on these issues, I cannot recommend this paper for publication in Nature Communications, since the programming scheme seems very difficult to accelerate, experimental accuracy is too far from software and there is no clear power analysis.

Point-to-point response:

REVIEWERS' COMMENTS:

Reviewer #1 (Remarks to the Author):

I think this manuscript is improved from the last version and I appreciate the authors responsiveness to my concerns about the neural network approach and the apples-to-apples comparison to alternative technologies.

The paper is well-suited for publication in my opinion. My one request would be that the authors include their answer to my comment 5a in the main text of the manuscript. Briefly, the authors look here at neural networks with fully-connected layers, and thus their potential benefit to convolutional and locally recurrent networks (such as LSTMs) are less obvious. That is fine for the state of this research and the finding (I can imagine a cleverly shaped memristor xbar configuration nicely suited for convolutions, for instance); however it is important that the general reader recognizes the difference. Most state-of-the-art neural network applications use these more sophisticated architectures; only a small number of the final layers in large networks like VGG or AlexNet are typically fully connected.

In addition, I think it may behoove the authors to mention that many of the algorithmic design choices for ANNs are determined in part by what is efficient on hardware. And if crossbars are fundamentally different than architectures that benefit from sparse matrix multiplications, that may change what configurations make sense (say, larger convolutional kernels?)

This does not diminish the impact of the paper, but does communicate its potential impact more appropriately to the broader readership of this journal.

-- Response: We sincerely thank the reviewer for the suggestions, and we agree that such analyses will boost the quality of our manuscript. In the final manuscript, we have added the following sentences on Page 12 to incorporate the additional discussion.

“It will be straightforward to build deeper fully-connected neural networks on an integrated chip with multiple large arrays in the near future, for even better accuracy and application to more complicated tasks. It is also noteworthy that most state-of-the-art deep neural networks involve sophisticated microstructures, e.g. convolutional neural networks (CNNs) or long short-term memory units (LSTMs). It may be worth investigating how to implement CNNs³⁷ or LSTMs efficiently on memristor crossbars in the future. But on the other hand, such microstructure-based algorithms have been developed for use on conventional hardware, on which it is more efficient to process sparse matrices. Since the advantages of using sparse matrices in a memristor crossbar are minimal, the optimal architectures for sophisticated tasks may look different.”

Reviewer #2 (Remarks to the Author):

The authors have answered all of my concerns and made a comprehensive revision.

Reviewer #3 (Remarks to the Author):

1 - I understand the authors' point of view regarding device programming speed, however they show 10 cycles at 5 ns, while the device would undergo millions of cycles during Neural Network Training, with endurance issues not clear yet.

-- Response: We agree that the endurance performance is very critical for the training of a memristor neural network. We have demonstrated that the same device can be switched for more than 120 billion cycles (far more than millions of cycles) even without the feedback circuits in the previous study (ref. 36), which is also reproduced here for your convenience.

Figure R1. 120 billion switching cycles have been demonstrated with pulses of 1.3 V/100 ns for SET and -3.05 V/100 ns for RESET. (Adapted from ref. 36).

2 - Authors state that x and Δ values are stored in software. I understand that this approach is easier for implementing the actual experiment, however it is not practical with the actual chip, since it would unacceptably slow down the overall operation. How the authors in practice think to implement this operation in chip? Saving the values in capacitors, with the risk of leakage, or using ADCs to store x and Δ in a near-located digital memory?

-- Response: We thank the reviewer for the comment. While we believe the focus of this paper is the demonstration of the feasibility of an emerging device based analog neural network by experiments rather than the circuit design, we also believe a complete implementation in an actual chip will become practical in the near future. Yes, it is true that we need to store the x values for each layer during the training, but the Δ values can be calculated by

backpropagation (as described in Page 7 in the main text) right before the weight update operation. The amount of data needs to be saved is proportional to the input dimension of the layer ($O(N)$), which is much less than that of the weight values ($O(N^2)$). The value can be saved either in capacitors or in an optimized sample and hold (S/H) circuit, depending on the required x accuracy given an expected weight update frequency for an application-specific scenario. We believe this will be an interesting research topic for future works.

3 - Programming scheme seems very slow (one row each time step). I fear this can be a conceptual problem, since this slows down the speed of training. My question to authors is: what is the possible commercial purpose of this memristor chip? To compete with GPUs or to focus on different applications? Authors should clearly state this, since they claim for “high speed-energy efficiency”. I fear it is not simple to overcome GPUs with row by row programming.

-- Response: Thank the reviewer for the comments. Although still at early stage, it is true that the possible commercial purpose of the memristor chip is to compete with other machine learning accelerators based on silicon technologies. Specifically for GPU, weight values are stored in DRAM, and the update of them is performed word line by word line, which is exactly the same as our proposed approach. But the memristor consumes less dynamic energy per update, and zero static energy due to its non-volatility. More importantly, other benefits of using memristor based computing system come from the fact that the computation takes place in-situ in the memory, removing the need to transfer the data between memory and computing unit entirely. The benefits of analog in-memory computing have been well established in previous literatures, as we discussed in Page 14.

4 - Fig. 4a is very useful, thank you for providing it. Experimental accuracy is around 5% below the corresponding TensorFlow accuracy, which seems a large gap, however.

-- Response: We thank the reviewer to acknowledge our effort in providing Fig. 4a. We would like to point out that the accuracy gap between our experiment and TensorFlow is 3.6% instead of 5% (and 2.4% between the experiment and a defect-free simulation). More importantly, our simulation on a larger memristor array (1024×512) with a defect rate of 11%, which is the same as that observed in our experiment, yields an accuracy of $97.3\% \pm 0.4\%$ on the MNIST classification test, approaching that demonstrated with traditional hardware.